# ProphetKV: User-Query-Driven Selective Recomputation for Efficient KV Cache Reuse in Retrieval-Augmented Generation

Shihao Wang [* 1]   Jiahao Chen [* 1]   Yanqi Pan [1]   Hao Huang [1]   Yichen Hao [1]   Xiangyu Zou [1]   Wen Xia [1]
Chongyang Qiu [2]   Wentao Zhang [2]   Pengfei Wang [2]

## Abstract

The prefill stage of long-context Retrieval-Augmented Generation (RAG) is severely bottlenecked by computational overhead. To mitigate this, recent methods assemble pre-calculated KV caches of retrieved RAG documents (by a *user query*) and reprocess selected tokens to recover cross-attention between these pre-calculated KV caches. However, we identify a fundamental "crowding-out effect" in current token selection criteria: globally salient but *user-query*-irrelevant tokens saturate the limited recomputation budget, displacing the tokens truly essential for answering the *user query* and degrading inference accuracy. We propose ProphetKV, a user-query-driven KV Cache reuse method for RAG scenarios. ProphetKV dynamically prioritizes tokens based on their semantic relevance to the *user query* and employs a dual-stage recomputation pipeline to fuse layer-wise attention metrics into a high-utility set. By ensuring the recomputation budget is dedicated to bridging the informational gap between retrieved context and the *user query*, ProphetKV achieves high-fidelity attention recovery with minimal overhead. Our extensive evaluation results show that ProphetKV retains 96%–101% of full-prefill accuracy with only a 20% recomputation ratio, while achieving accuracy improvements of 8.8%–24.9% on RULER and 18.6%–50.9% on LongBench over the state-of-the-art approaches (e.g., CacheBlend, EPIC, and KVShare).

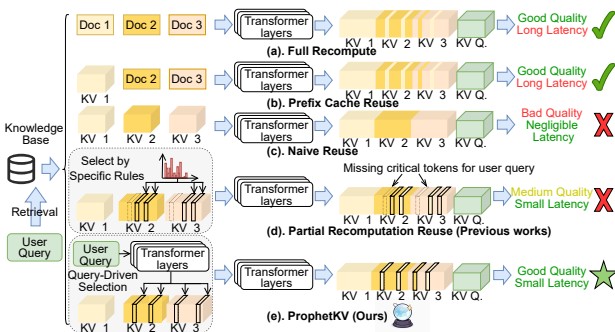

*Figure 1.* A high-level overview of different methods of KV Cache reuse in RAG scenarios.

## 1. Introduction

Large Language Models (LLMs) integrated with Retrieval-Augmented Generation (RAG) have become the *de facto* standard for addressing domain-specific tasks (Gao et al., 2023; Chang et al., 2025; Zhao et al., 2026). In a typical RAG pipeline, the system retrieves a collection of relevant document chunks from a vast external corpus based on a *user query*, which are then concatenated to form the input context. However, to ensure comprehensive grounding and high answer quality, modern RAG systems often need to process an increasing number of retrieved fragments, scaling the total context length from 10k to 1M tokens (Jiang et al., 2024; Ataallah et al., 2025; Yao et al., 2025).

This massive input size imposes a severe computational burden, particularly during the prefill stage of LLM inference. While LLM inference is generally divided into a prefill stage (processing the prompt) and a decode stage (generating tokens), the former becomes the primary bottleneck in long-context RAG scenarios. Unlike the autoregressive decode stage, the prefill stage must compute self-attention across the entire sequence to aggregate semantic information into a Key-Value (KV) cache. As the computational complexity of the attention mechanism scales as $O(N^2)$ with sequence length $N$, the prefill latency (i.e., Time-to-First-Token, TTFT) becomes prohibitively high, severely hindering the responsiveness of real-time RAG services.

To alleviate the prefill bottleneck, KV cache reuse has emerged as a cornerstone strategy. Traditional methods

---
[*]Equal contribution   [1]Harbin Institute of Technology, Shenzhen [2]Beijing Yanrong Technology Co., Ltd..   Corresponding authors: Xiangyu Zou <zouxiangyu@hit.edu.cn>, Wen Xia <xiawen@hit.edu.cn>.

*Proceedings of the 43rd International Conference on Machine Learning*, Seoul, South Korea. PMLR 306, 2026. Copyright 2026 by the author(s).

rely on strict prefix matching (Zheng et al., 2024), which constrains reuse to identical sequences. However, this requirement is rarely satisfied in RAG scenarios, where retrieved documents are dynamically reordered and seldom share a common prefix (Yao et al., 2025). Consequently, recent research has shifted toward position-independent (PI)[1] KV cache reuse (Yao et al., 2025; Hu et al., 2024), enabling the assembly of precomputed chunk-wise caches regardless of their original sequence order.

However, simply concatenating KV caches that were precomputed in isolation causes severe accuracy degradation. This is because such an approach neglects **cross-attention** between documents: they have never "seen" each other during precalculation. To reinstate these inter-document dependencies, state-of-the-art methods (Yao et al., 2025; Hu et al., 2024) introduce partial recomputation, selectively recomputing the KV cache for a small subset of tokens to "bridge" the resulting semantic fragmentation and strike a balance between computational savings and inference accuracy.

Despite these efforts, we identify a fundamental flaw in existing recomputation schemes: their selection criteria are inherently blind. By relying on global attention weights or positional heuristics, these methods strive to approximate the full cross-attention map of a standard Transformer prefill. However, they fail to distinguish between generic saliency and task-specific relevance: only a sparse subset of pivotal sentences is task-critical in retrieved documents, whereas the vast other content remains semantically extraneous to the specific *user query*. Reconstruction cross-attention for these "unused" tokens provides marginal utility for final answer generation but incurs a substantial recompute cost. This induces a "crowding-out effect": the limited recomputation budget is saturated by globally active but task-irrelevant tokens, thereby displacing the tokens truly essential for accurate answer generation. As a result, we observe that existing methods suffer accuracy drops of up to 86% on representative benchmarks (See Sec. 5.2), which limits their viability for real-world deployments and motivates the design of higher-fidelity recomputation methods.

Accordingly, we encapsulate our findings into **two core insights**: (1) The reconstruction objective adopted by prior methods is functionally redundant for RAG tasks; (2) The utility of cross-attention is strictly query-contingent. In RAG scenarios, the user query serves as a decisive semantic prior that defines the tokens' relevance.

These insights motivate a user-query-targeted selector, but applying such a selector inside a Transformer still raises a

key design question: token importance is not stable across layers. Previous proposed selection schemes mainly rely on a single layer approximation to avoid the cost of full-depth scoring which can poorly match the tokens preferred by deeper layers in our observation (Fig. 7). As depth increases, attention shifts from local lexical cues to higher-level evidence aggregation, so a token that appears unimportant in one layer may become essential in another.

Based on these considerations, we propose ProphetKV, a user query-driven selective recomputation framework (Fig. 1). ProphetKV facilitates a paradigm shift from blind approximation of the global attention landscape to query-targeted attention recovery, ensuring that partial recomputation mechanisms are dedicated exclusively to repairing cross-attention relevant to the specific query. ProphetKV optimizes the accuracy-efficiency trade-off through two synergistic components: **(1) User Query-Guided Token Selection**: We introduce a mechanism that leverages the attention weights of the user query to dynamically isolate context tokens that are semantically pivotal for the user query. **(2) Dual-Stage Recomputation with Layer Fusing**: Building on the cross-layer issue above, we first collect query-aware importance signals layer by layer and then fuse them before selecting the recomputation set. This design makes inter-layer attention variance an explicit part of token selection, yielding a unified, high-utility recomputation set.

Experimental results in Sec. 5.2 demonstrate that, under a constrained budget of 20% recomputed tokens, our method achieves significant accuracy gains of 8.8-24.9% on the RULER dataset and 18.6-50.9% on the LongBench dataset over prior SOTA approaches. Notably, our approach is the only strategy capable of nearing full recomputation accuracy, underscoring its potential to improve the accuracy-efficiency trade-off in long-context RAG.

**Conflict of Interest Disclosure.** This work was conducted while authors Shihao Wang and Jiahao Chen were interns at Beijing Yanrong Technology Co., Ltd. The company provided research support and computational resources for this work. The experiments in this paper do not evaluate any proprietary models, products, or serving systems developed by the company.

## 2. Background

### 2.1. KV Cache in the Transformer architecture

Transformers aggregate contextual information via causal self-attention, where each token is associated with Query (Q), Key (K), and Value (V) representations. During inference, the model initiates a prefill stage to process the input prompt, caching the resulting Key and Value tensors (KV cache), which are reused to facilitate efficient token generation in the subsequent decoding stage. The model

---

[1]To ensure position independence, positional encodings are excluded from precomputed Key caches and restored at runtime (Yao et al., 2025). We thus treat this mechanism as given and focus on the reuse strategy.

is organized as a stack of layers, where the output of each layer serves as the input to the next, forming a hierarchical dependency that captures long-range and abstract semantics.

## 2.2. KV Cache Reuse and RAG applications

RAG augments LLMs with externally retrieved document chunks and prepends them to the input prompt to guide generation, resulting in long input sequences. We discuss prompt-layout robustness for this context-before-query assembly in Appendix B.1. Since retrieved chunks are frequently identical across requests, reusing their KV caches offers a promising opportunity to reduce prefill latency. Existing inference systems such as vLLM (Kwon et al., 2023) and SGLang (Zheng et al., 2024) employ prefix-based KV cache reuse, which reuses KV caches only when two requests share an identical prompt prefix. However, this strict requirement is ill-suited for RAG: even if two requests share identical chunks, any variation in their ordering breaks the prefix chain, rendering the KV cache non-reusable.

To address this limitation, position-independent (PI) KV cache reuse decouples precomputed KV caches from their absolute token positions. A naïve PI reuse strategy directly concatenates precomputed chunk caches, which maximizes computational savings but degrades accuracy due to missing cross-attention interactions. Recent work alleviates this issue via partial recomputation, which selectively recomputes a small subset of tokens to reconstruct cross-attention.

Existing methods fall into two categories: **training-free** and **fine-tuned** approaches. Training-free methods are classified as static or dynamic, depending on whether token selection depends on the input prompt. EPIC (Hu et al., 2024) is a typical static method based on the attention-sink phenomenon to select the initial tokens of each chunk for recomputation[2]. Methods based on dynamic rules include CacheBlend (Yao et al., 2025) and KVShare (Yang et al., 2025c); these derive token-selection rules from numerical error analyses of deviations in the KV cache and in hidden states, respectively. Fine-tuned methods are represented by CacheClip (Yang et al., 2025b), which fine-tunes a small auxiliary model to predict recomputation-worthy tokens by exploiting similarity between the auxiliary and target models. While effective under controlled settings, such approaches suffer from heavy run-time overhead and limited practicality in open-domain or rapidly evolving workloads.

Given the practical limitations of fine-tuned approaches, this work focuses on training-free, plug-and-play partial recomputation for KV cache reuse. We do not consider finetuned solutions, as our goal is to preserve the native generalization of LLMs and avoid additional training. Nevertheless, exist-

ing training-free partial recomputation methods still struggle to achieve a satisfactory balance between accuracy and computational efficiency on various benchmarks (see Sec. 5), indicating substantial room for further improvement.

## 3. Motivation

### 3.1. Illustrating the Failure of Existing Methods

Existing partial-recomputation methods aim to reconstruct the entire missing cross-chunk attention under the assumption that this is feasible within a strict budget. However, our evaluation in Sec. 5.2 reveals a consistent failure to maintain accuracy in various RAG scenarios.

We analyze a representative RAG case in which a query, "In which city does Alice stay on Monday?" requires bridging information between Chunks 1 and 3, whereas Chunk 2 contains misleading details. As shown in Fig. 2, all existing methods fail to reconstruct this significant cross-attention and thus predict an incorrect answer, i.e. "Paris". Notably, the selected tokens of these methods do not align with user-query-relevant tokens identified by human intuition. We quantify this mismatch by measuring the overlap ratio, an indicator widely used in prior works (Yang et al., 2024; 2025b), to evaluate the consistency between selected tokens and query-critical tokens in Fig. 3. The combined results of Fig. 2 and Fig. 3 suggest that **the impact of missed user query-related tokens leads to incorrect predictions**.

**Degradation of Current Recomputation Methods.** EPIC relies on static heuristics, which lack the dynamic, input-dependent flexibility of Transformers. CacheBlend and KVShare utilize deviation-based criteria that, while theoretically motivated, are difficult to estimate without a full prefill pass. To reduce overhead, they approximate high-layer dependencies using low-layer information. However, since low-layer representations are often misaligned with high-layer semantics, these methods fail to capture tokens whose relevance only emerges in deeper layers(See Sec. 4.3). As a natural result, these methods fail to recover the whole missing cross-attention, resulting in the failure in Fig. 2.

This reveals a fundamental flaw in the goal of existing methods: by trying to recover all missing cross-attention within limited budgets, they saturate the available budget with irrelevant information, crowding out the tokens essential for query accuracy. This evidence casts doubt on whether such an objective is overly ambitious. From a structural standpoint, if a lightweight mechanism could faithfully recover full cross-attention semantics, it would supplant standard Transformer attention, as the combined cost of precomputation and reconstruction would be significantly lower than the full prefill cost [3]. However, current methods show no

---

[2]However, attention-sink effects can also be mitigated by multiple recomputation strategies implicitly, see Appendix D.1.

[3]For a sequence of length $s$ equally partitioned into $N$

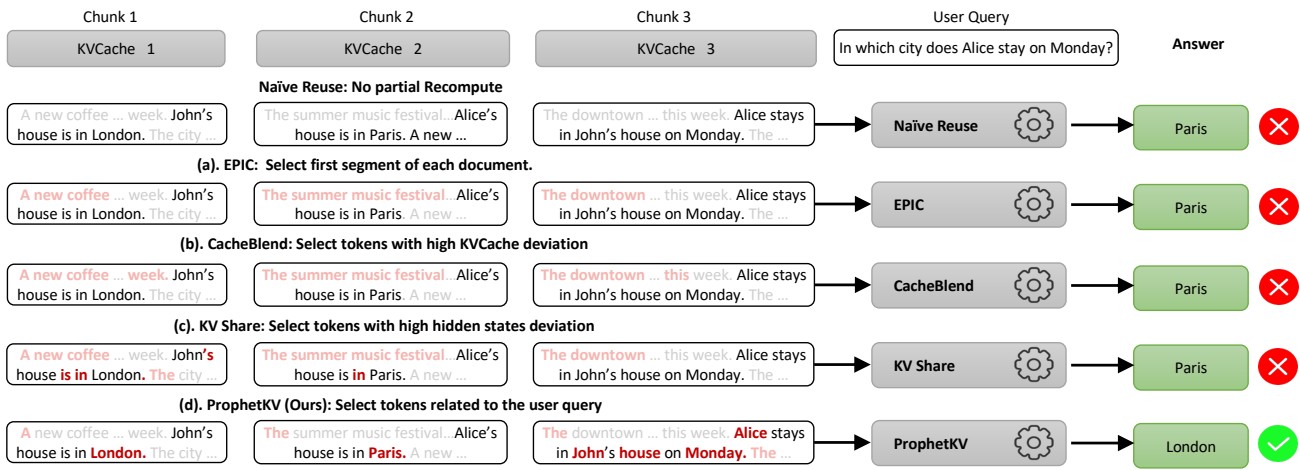

*Figure 2.* An example illustrating the selected tokens of existing approaches under a 20% recomputation ratio. Text irrelevant to the user query is colored in gray, and tokens selected by each method are colored in red. See Appendix B for the full prompt.

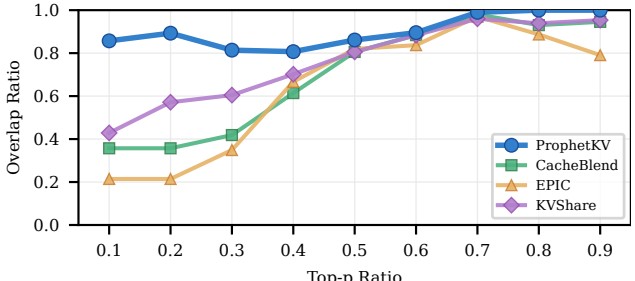

*Figure 3.* Overlap ratio ($|S \cap G|/|G|$) between selected tokens ($S$) and query-attended tokens ($G$), where $G$ is the average query-to-context attention under full-prefill, across selection ratios ($p$).

such potential; instead, they behave as degraded versions of the original Transformer, reflecting an inevitable trade-off under limited computational resources.

### 3.2. Our Insight: The User Query as a Prophet

The above analysis indicates that the missing cross-attention cannot be fully reconstructed in most cases, suggesting that the objective of global attention recovery should be replaced with a more targeted approach. We contend that cross-attention utility is inherently query-contingent: it serves as the bridge between the user's intent and the retrieved evidence. Therefore, recovering attention for query-irrelevant tokens provides marginal utility for the final answer.

While identifying task-relevant intent is typically difficult due to the non-unified representation of queries, RAG systems offer a distinct structural invariant: the query is almost always placed at the prompt's conclusion. We capitalize on this layout as an opportunity: by treating the terminal query as a 'prophet,' we can extract precise relevance signals to guide selective recomputation. This transforms a structural convention into a powerful mechanism for high-fidelity attention recovery with minimal cost.

equal chunks, the complexity of chunk-wise precomputation is $\mathcal{O}(s^2/N)$, compared to $\mathcal{O}(s^2)$ for full attention.

**The predictive power of query-context attention.** As illustrated in Fig. 4, the subsets attended by the user query exhibit a consistently high overlap ratio with those attended during decoding, a property that remains robust across various models. This observation suggests that the user query acts as a *prophet*, revealing which parts of the document are critical for the upcoming generation. Importantly, this signal is available at every Transformer layer during the lightweight query pass. This is crucial because single-layer approximations, which are commonly adopted by prior efficient selection methods to control overhead, provide only a partial view of token importance and can deviate substantially from the layer-wise optimal subsets (Fig. 7). ProphetKV therefore treats query attention as a layer-wise signal and later fuses these signals into a stable recomputation set, rather than committing to a fixed layer prematurely.

**Advantages.** ❶ This property enables targeted recomputation of high-priority tokens guided by the query's own foresight, rather than blindly reconstructing the entire missing cross-attention, thereby improving generation quality (Sec. 4.2). ❷ It also provides token-importance signals across all layers without a full prefill, which makes layer fusion practical and avoids the computational deadlock observed in prior work (Sec. 4.3).

## 4. Design

### 4.1. Overview of ProphetKV

Motivated by this insight, we propose ProphetKV, a high-fidelity, position-independent KV cache reuse mechanism based on query-driven selective recomputation. As shown in Fig. 5, it employs a dual-stage framework: Stage I generates an evaluation metric based on the user query, and Stage II uses this metric to guide selective recomputation. Implementing this approach, however, poses two key challenges:

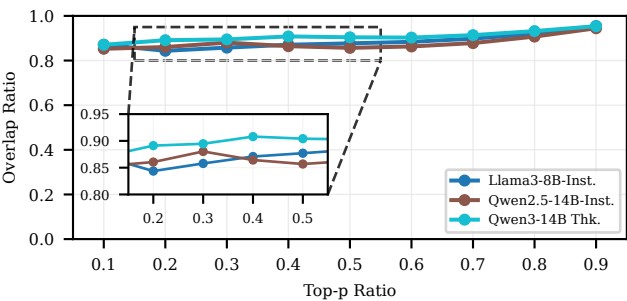

*Figure 4.* The overlap ratio between query-attended tokens and actual critical tokens in the decoding stage. Across various model families, query attention consistently predicts the tokens that will be attended during the decoding stage.

**Challenge 1: How to quantify token importance via query attention?** We need a quantitative metric for token selection that accurately evaluates the query's attention focus while remaining computationally efficient.

**Challenge 2: How to handle attention variability across Transformer layers?** Attention patterns vary significantly across layers to capture diverse semantic features. This variability makes it difficult to design a selection mechanism that remains robust across the entire depth of LLMs.

### 4.2. Query-Driven Token Importance Quantification

For **Challenge 1**, we formalize the relationship between query attention and the fidelity of the generated output, bridging the gap between intuition and mathematical rigor.

**Define the numerical loss function.** Let $s$ be the number of input tokens, where indices $1 \leq t \leq s$ and $t > s$ denote input and generated tokens, respectively. Let $Q_s$ be the set of user query tokens. We denote $V_t$ as the $t$-th token's Value tensor and $\Phi_{n,t}$ as the attention weights of the token $n$ to $t$; e.g., $\Phi_{Q_s,t}$ means the attention weight of user query tokens to the $t$-th token. For any generated token $n > s$, the output of the attention module can be decomposed as

$$\text{AttnOut}_n = \sum_{t \leq n} \Phi_{n,t} V_t = \sum_{1 \leq t \leq s} \Phi_{n,t} V_t + \sum_{s < t \leq n} \Phi_{n,t} V_t. \quad (1)$$

The second term $\sum_{s<t\leq n} \Phi_{n,t} V_t$ corresponds to the contribution of previously generated tokens, and in this work, we focus on the first term, $\sum_{1 \leq t \leq s} \Phi_{n,t} V_t$, which captures the impact of the input tokens (i.e., reused KV caches). Guided by insights from Fig. 4, we recognize that the average attention weights of the query tokens, denoted by $\hat{\Phi}_{Q_s,t} = \frac{1}{|Q_s|} \sum_{q \in Q_s} \Phi_{q,t}$, serve as a reliable proxy for $\Phi_{n,t}$. This motivates us to leverage the approximation $\Phi_{n,t} \approx \hat{\Phi}_{Q_s,t}$ (for $n > s$) as a heuristic to derive a quantitative metric for evaluating token importance.

Next, we define the *semantic loss* arising from position-independent KV reuse. In this scenario, K and V tensors are computed for each document in isolation. Such KV caches lack the global context typically provided by cross-document attention, causing both $V_t$ and $\hat{\Phi}_{Q_s,t}$ to become imprecise. We denote these imprecise counterparts as $V_t'$

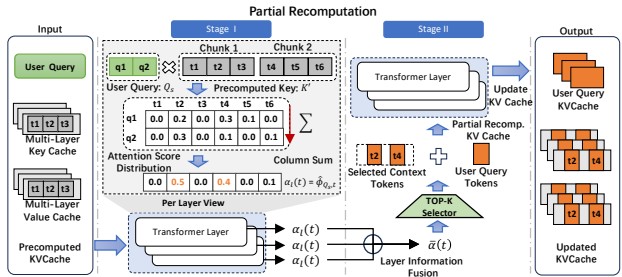

*Figure 5.* Overview of the proposed method: utilizing the query as a prophet to guide selective KV cache recomputation.

($V_t$ without cross-attention) and $\hat{\Phi}'_{Q_s,t}$ (average attention weights of query tokens to imprecise Key caches), and formulate the resulting semantic loss as follows:

$$\mathcal{L}_{\text{semantic}} = \left\| \sum_{1 \leq t \leq s} \hat{\Phi}_{Q_s,t} V_t - \sum_{1 \leq t \leq s} \hat{\Phi}'_{Q_s,t} V_t' \right\|_2. \quad (2)$$

Directly optimizing Eq. 2 is computationally intractable because the losses across tokens are coupled. Following prior practices (Willette et al., 2025; Liu et al., 2023), we derive a tractable upper bound using the triangle inequality:

$$\mathcal{L}_{\text{semantic}} \leq \mathcal{L} = \sum_{1 \leq t \leq s} \left\| \hat{\Phi}'_{Q_s,t} V_t' - \hat{\Phi}_{Q_s,t} V_t \right\|_2 \quad (3)$$

**Derive the ideal and practical value functions.** To mitigate the loss defined in Eq. 3 through partial recomputation, we prioritize recomputing KV caches for only some "critical" tokens. Following prior studies (Yao et al., 2025; Yang et al., 2025c;b), we assume that recomputation restores the KV caches of selected tokens to their ground-truth values ($\hat{\Phi}_{Q_s,t}$ and $V_t$) with negligible numerical errors.

Let $\alpha(t)$ be a value function used to identify and rank critical tokens for recomputation. Given a recomputation budget of $p$ ratio, the residual loss incurred by the unrecomputed tokens can be formulated as:

$$\mathcal{L}_{\alpha,p} = \sum_{t \notin \text{TOP}_p(\alpha(t))} \left\| \hat{\Phi}'_{Q_s,t} V_t' - \hat{\Phi}_{Q_s,t} V_t \right\|_2, \quad (4)$$

where $\text{TOP}_p(\alpha(t))$ denotes the top $p \in [0, 1]$ fraction of tokens ranked by $\alpha(t)$.

Our aim is to minimize $\mathcal{L}_{\alpha,p}$ through a well-designed $\alpha(t)$.

Obviously, one of the ideal value functions is as follows:

$$\alpha_{\text{ideal}}(t) = \left\| \hat{\Phi}_{Q_s,t} V_t - \hat{\Phi}'_{Q_s,t} V_t' \right\|_2, \quad (5)$$

because it directly yields the loss for each token to identify "critical" tokens, although it cannot be computed in a simple manner. To derive its tractable formula, we conduct:

$$\alpha_{\text{ideal}}(t) = \left\| (\Delta\hat{\Phi}_{Q_s,t}) V_t' + (\hat{\Phi}'_{Q_s,t} + \Delta\hat{\Phi}_{Q_s,t}) \Delta V_t \right\|_2, \quad (6)$$

where $\Delta\hat{\Phi}_{Q_s,t} = \hat{\Phi}_{Q_s,t} - \hat{\Phi}'_{Q_s,t}$ and $\Delta V_t = V_t - V_t'$.

Eq. 6 identifies four factors influencing recomputation priority: $|\Delta\hat{\Phi}_{Q_s,t}|$, $||V_t'||_2$, $\hat{\Phi}'_{Q_s,t}$, and $||\Delta V_t||_2$. To quantify

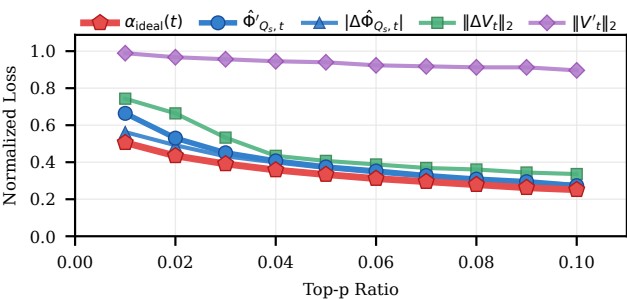

*Figure 6.* Comparison of token selection criteria for minimizing semantic loss. $\hat{\Phi}'_{Q_s,t}$ well approximates the ideal value function while remaining observable without a full context prefill.

the individual contribution of each one, we conduct an empirical sensitivity analysis by setting $\alpha(t)$ to each candidate in turn and measuring the resulting residual loss $\mathcal{L}_{\alpha,p}$. The results are shown in Fig. 6, $\alpha(t) = |\Delta\hat{\Phi}_{Q_s,t}|$ yields the lowest residual loss, followed closely by $\hat{\Phi}'_{Q_s,t}$ and $||\Delta V_t||_2$, while $||V'_t||_2$ performs poorly.

However, $|\Delta\hat{\Phi}_{Q_s,t}|$ and $||\Delta V_t||_2$ are only obtainable after recomputation, making them unsuitable as recomputation criteria. Conversely, $\hat{\Phi}'_{Q_s,t}$ is directly observable by a lightweight process that runs Transformers only on the short user query with the imprecise KV caches. Given this computation convenience, we adopt $\alpha(t) = \hat{\Phi}'_{Q_s,t}$ as our practical proxy for the ideal value function.

### 4.3. Overcoming the Deadlock in Token Selection

Applying selection criteria within the Transformer architecture presents a fundamental challenge (**Challenge 2**) for prior approaches. These approaches (Yao et al., 2025; Yang et al., 2025c) primarily rely on the magnitude of KV tensors to identify critical tokens. However, this strategy is hindered by the layer-wise visibility constraint of Transformers: the KV magnitudes at layer $l$ can only be computed after the full computation of layer $l-1$ is complete.

This creates a structural computational deadlock: to accurately assess token importance at each layer for selective recomputation, these methods would require a full forward pass through all layers. Paradoxically, this amounts to a complete prefill of the entire context—the very computation that KV reuse is intended to avoid—thereby rendering the reuse mechanism ineffective. To circumvent this issue, prior work often assumes layer-wise similarity, namely that tokens deemed important in early layers remain equally important throughout the model. However, this assumption contradicts the fundamental design of Transformers, in which different layers specialize in capturing distinct semantic features at varying levels of abstraction. Consequently, these methods often struggle to balance selection accuracy with system throughput.

Fortunately, our proposed selection criterion $\hat{\Phi}'_{Q_s,t}$ is in-

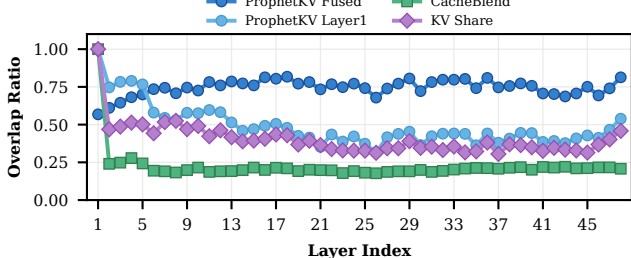

*Figure 7.* Study on the effectiveness of our fused value function. We evaluate the overlap ratio between the actually selected tokens in each methods and the layer-wise optimal tokens selected by different methods (Appendix C.1). ProphetKV Layer1, CacheBlend, and KVShare select token only depends on the first layer, whereas ProphetKV Fused considers all layers in token selections.

herently immune to this deadlock. Specifically, $\hat{\Phi}'_{Q_s,t}$ represents the attention weights from the query tokens to the document KV caches across all layers, which provides token importance in all layers without requiring a full prefill.

To ensure robust generalizability (See Appendix C.4), we fuse these per-layer insights into a single global value function. Notably, since attention weights are inherently normalized within each Transformer layer, they provide a consistent scale for comparison across layers. Leveraging this property, we adopt a uniform fusion strategy to avoid the brittleness of manual layer selection as:

$$\bar{\alpha}(t) = \frac{1}{L}\sum_{l=1}^{L}\alpha_l(t). \quad (7)$$

As shown in Fig. 7, the fused value function achieves much higher overlap with the layer-wise optimal subset, indicating that it reliably captures critical tokens across layers. This choice is also supported by the fixed-depth ablation in Appendix Tab. 6: different tasks favor different depths, while average fusion remains competitive with the best fixed-depth choices and avoids committing to a brittle layer.

Note that our proposed selection criterion $\hat{\Phi}'_{Q_s,t}$ preserves a per-layer computational cost of $\mathcal{O}(|Q_s| \times s)$, which is lower than the $\mathcal{O}(s^2)$ cost required by CacheBlend or KVShare (Appendix C.1) in the common case where $|Q_s| \ll s$.

**Putting It Together: The Dual-Stage Recomputation Pipeline. Stage I:** We perform a "lightweight pass" across all layers. Instead of full attention, we only compute the attention weights of the current query tokens $Q_s$ to the context tokens. For each layer $l$, we use the column sums of these attention weights to calculate the value function $\hat{\Phi}'_{Q_s,t}$. **Stage II:** Then, we compute the fused value function $\bar{\alpha}(t)$ as Eq. 7 and select the top-$k$ most critical tokens. We then perform a complete recomputation across all layers, but only for these selected tokens. Although one could select a different token set for each layer, we keep a unified set to preserve regular tensor shapes, simplify cache replacement, and avoid extra dependencies between selection and recomputation. This makes the recomputation path more favorable for GPU throughput and kernel utilization.

*Table 1.* Performance comparison of five approaches on RULER (left side of the table) and LongBench (right side of the table) across various models under a 20% recomputation ratio. Results at other context lengths are provided in Appendix F.1. The complete datasets for all LongBench tasks are presented in Appendix F.2.

| Methods | CWE | FWE | MK1 | MQ | MV | Single | QA1 | QA2 | VT | Avg. | WQA | TQA | HQA | NQA | MQue | QMSum | PRetr_en | PRetr_zh | Avg. |
|---|---|---|---|---|---|---|---|---|---|---|---|---|---|---|---|---|---|---|---|
| *Llama3.1-8B-Inst.* | 96.90 | 93.67 | 100.00 | 98.75 | 100.00 | 100.00 | 77.42 | 48.00 | 95.80 | 88.82 | 43.14 | 43.31 | 53.35 | 25.50 | 26.20 | 22.19 | 99.67 | 88.18 | 50.19 |
| NaiveReuse | 93.00 | 90.33 | 62.00 | 54.75 | 32.00 | 72.00 | 64.42 | 46.00 | 38.40 | 61.36 | 34.88 | 44.31 | 42.92 | **24.48** | 17.25 | 21.53 | 39.33 | 14.00 | 29.84 |
| CacheBlend | 93.60 | 89.33 | 77.00 | 82.25 | 73.50 | **98.67** | 77.17 | 49.00 | 54.60 | 77.27 | 39.10 | **44.79** | 46.75 | 22.77 | 20.91 | 22.60 | 73.67 | 48.00 | 39.82 |
| EPIC | **96.10** | **93.00** | 76.00 | 67.00 | 50.50 | 97.00 | 70.75 | **52.00** | 36.20 | 70.32 | 40.39 | 44.65 | 47.46 | 24.03 | 17.88 | 23.01 | 65.33 | 20.50 | 35.41 |
| KVShare | 92.90 | 87.33 | 73.00 | 79.50 | 67.25 | 97.00 | 67.42 | 45.00 | 51.40 | 73.47 | 43.85 | 45.48 | | 23.55 | 20.50 | 22.76 | 65.67 | 51.00 | 38.67 |
| ProphetKV | 95.50 | 90.67 | 99.00 | 98.50 | 100.00 | 97.67 | **77.33** | 51.00 | 67.00 | **84.71** | 43.21 | 44.12 | 50.69 | 23.68 | 24.38 | 23.35 | 99.00 | 98.00 | **50.80** |
| *Qwen2.5-14B-Inst.* | 97.70 | 94.00 | 100.00 | 99.75 | 98.50 | 100.00 | 68.92 | 64.00 | 99.40 | 90.28 | 54.45 | 40.57 | 62.11 | 26.78 | 33.56 | 22.94 | 99.67 | 98.79 | 54.86 |
| NaiveReuse | 94.00 | 98.00 | 69.00 | 55.75 | 29.25 | 96.00 | 51.00 | 43.00 | 35.60 | 62.83 | 17.82 | 9.84 | 24.40 | 15.24 | 2.95 | 19.84 | 28.03 | 8.72 | 15.86 |
| CacheBlend | 96.40 | 99.00 | 94.00 | 93.00 | 59.25 | **99.67** | 60.00 | 59.00 | 58.60 | 78.11 | 44.39 | 27.42 | 52.61 | 22.25 | 27.02 | 21.48 | 69.11 | 18.92 | 35.40 |
| EPIC | **97.30** | 98.67 | 88.00 | 91.50 | 45.25 | 98.33 | 64.67 | 55.00 | 41.60 | 74.04 | 41.70 | 22.94 | 53.36 | 24.48 | 24.00 | 21.98 | 45.72 | 13.76 | 30.99 |
| KVShare | 95.80 | 98.67 | 92.00 | 88.50 | 48.75 | **99.67** | 63.00 | 53.00 | 39.40 | 73.35 | 43.85 | 25.80 | 50.07 | 23.93 | 26.76 | 21.76 | 65.06 | 21.30 | 34.82 |
| ProphetKV | 96.00 | **99.33** | 97.00 | 97.00 | 91.25 | **99.67** | 70.17 | 60.00 | 95.40 | **88.60** | 52.32 | 39.82 | 58.13 | 26.68 | 35.60 | 22.51 | 99.67 | 92.67 | **53.43** |
| *Qwen3-14B-Thk.* | - | - | 100.00 | 100.00 | 98.00 | 100.00 | 78.75 | 74.00 | 100.00 | 91.79 | 73.01 | 46.16 | 74.24 | 25.43 | 49.10 | 20.58 | 100.00 | 100.00 | 61.06 |
| NaiveReuse | - | - | 50.00 | 49.25 | 27.75 | 75.00 | 54.33 | 41.00 | 15.80 | 43.85 | 25.31 | 37.77 | 26.08 | 7.62 | 3.88 | 5.18 | 24.65 | 8.50 | 17.37 |
| CacheBlend | - | - | 72.00 | 77.50 | 57.25 | 98.33 | 70.08 | 64.00 | 64.80 | 71.99 | 63.91 | 45.30 | 63.85 | 23.82 | 37.81 | **19.24** | 73.90 | 73.50 | 50.17 |
| EPIC | - | - | 68.00 | 75.25 | 51.50 | 99.00 | 71.08 | 67.00 | 43.80 | 67.94 | 62.81 | 45.19 | 61.61 | 22.51 | 37.79 | 18.96 | 66.00 | 47.00 | 45.23 |
| KVShare | - | - | 68.00 | 77.25 | 56.50 | **99.33** | 72.75 | 65.00 | 53.00 | 70.64 | 61.84 | 45.26 | 64.95 | 22.61 | 33.26 | 18.49 | 71.78 | 73.00 | 48.90 |
| ProphetKV | - | - | **97.00** | 96.50 | 95.75 | 98.67 | **76.42** | 72.00 | 100.00 | **89.89** | 70.83 | 46.53 | 70.77 | 25.33 | 44.12 | 19.07 | 100.00 | 99.50 | **59.52** |

Note: For CWE and FWE with Qwen3-14B-Thk., the model repeats the entire input tokens during generation, exceeding the maximum output length; therefore, these results are excluded.

# 5. Evaluation

We evaluate ProphetKV's ability to optimize the accuracy-efficiency trade-off by addressing the following questions: (i) **Accuracy**: Can ProphetKV maintain high accuracy under an aggressive recomputation ratio (§5.2)? (ii) **Efficiency**: What are the practical latency gains in real-world long-context settings (§5.3)? (iii) **Robustness**: How does the method generalize across varying configurations (§5.4)?

## 5.1. Environment Setup

**Models and Hardware.** We evaluate ProphetKV on three representative LLMs: **Llama3.1-8B-Instruct** (Grattafiori et al., 2024), **Qwen2.5-14B-Instruct** (Team et al., 2024), and **Qwen3-14B** (Yang et al., 2025a). To accommodate the long-chain reasoning capabilities of Qwen3-14B, we enable its thinking mode and set the maximum output length to 4K tokens (Wang & Zhou, 2024; Lin et al., 2025); for the other models, we follow the standard limits established in prior work (Cai et al., 2024). Accuracy experiments are conducted on a heterogeneous cluster equipped with NVIDIA A100, H100, and L20 GPUs.

**Benchmarks.** We employ two widely used benchmark suites: **RULER** (Hsieh et al., 2024) (8K context length; see Appendix F.1 for results with extended lengths) for retrieval-intensive stress testing, and **LongBench** (Bai et al., 2024) for reasoning and summarization tasks. Both datasets are partitioned into 512-token segments using LangChain (LangChain, 2024).

**Baselines.** We compare ProphetKV with three state-of-the-art methods: **CacheBlend** (Yao et al., 2025), **KVShare** (Yang et al., 2025c), and **EPIC** (Hu et al., 2024). Additionally, a **Naive Reuse** baseline is included to define

the lower bound, as it does not perform any recomputation.

**Implementation.** We implement ProphetKV and all baselines using the HuggingFace Transformers framework (Wolf et al., 2019). This setup isolates algorithmic performance from effects introduced by specific CUDA kernels (e.g., FlashAttention (Dao, 2023), FlexAttention (Dong et al., 2024)). Such system-level optimizations are orthogonal to our method and can be incorporated for additional gains. We use *Time to First Token* (TTFT) as the primary metric to capture both prefill and recomputation latency.

## 5.2. Accuracy Evaluation

**Performance Overview.** ProphetKV achieves accuracy comparable to full recomputation (Tab. 1). Naive Reuse suffers an accuracy degradation due to the loss of cross-chunk information, and existing baselines exhibit unstable performance across tasks. In contrast, ProphetKV identifies tokens influential for generation. This advantage is pronounced in tasks requiring precise localization of relevant spans under contextual interference, such as multi-value (MV) and multi-query (MQ) tasks. Under a constrained recomputation budget of 20% tokens, ProphetKV achieves an accuracy improvement of 8.8%-24.9% on RULER and 18.6%-50.9% on LongBench over prior state-of-the-art methods.

**Cross-Dataset and Cross-Model Robustness.** Across both RULER and LongBench, ProphetKV outperforms all other baselines. On LongBench, selectively recomputing a small subset of query-relevant tokens preserves long-range semantic coherence even beyond 16K tokens. Moreover, the advantage remains consistent across model scales (8B to 14B) and types (instruction-tuned vs. reasoning-oriented), demonstrating that the query-driven mechanism generalizes

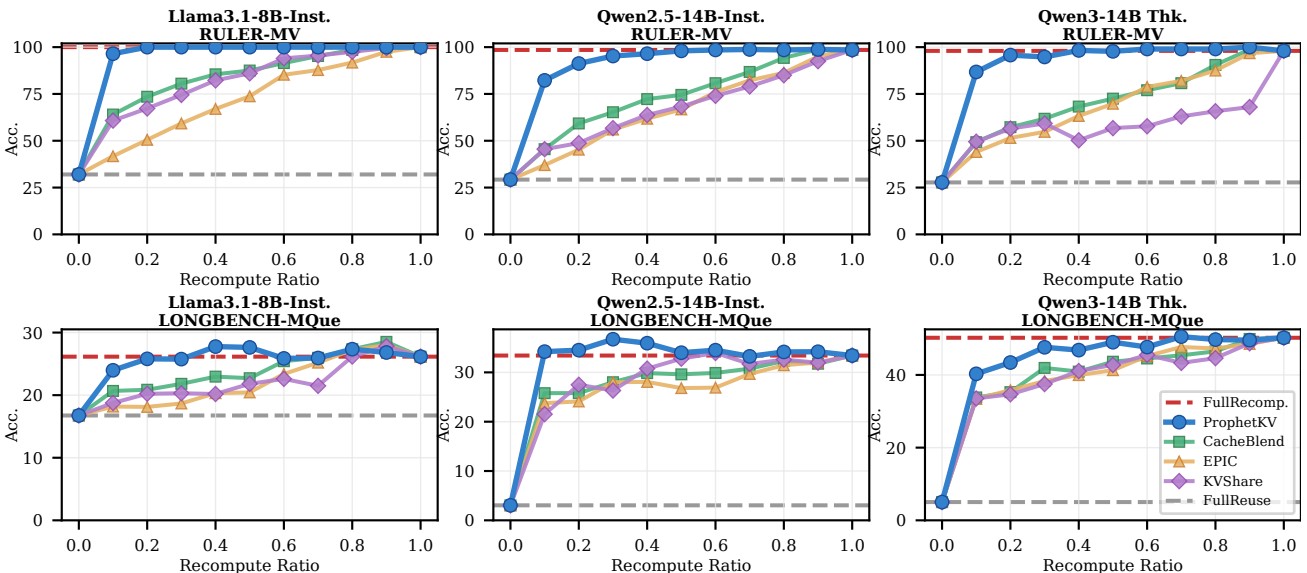

*Figure 8.* Accuracy results on RULER MultiValue and LongBench Musique across different models and recomputation ratios.

*Table 2.* TTFT results across different models and context lengths. Each cell shows TTFT in seconds, measured with batch size 1 on a single H100 GPU. Complete results, including all baselines, are provided in Appendix E.3.

| Model | Context | FullRecomp. | EPIC | ProphetKV | CacheBlend |
|---|---|---|---|---|---|
| Llama3.1-8B-Inst. | 16K | 5.23 | 1.08 | **1.13** | 1.29 |
| | 8K | 1.48 | 0.32 | **0.35** | 0.38 |
| | 4K | 0.46 | 0.14 | **0.15** | 0.14 |
| Qwen2.5-14B-Inst. | 16K | 9.94 | 2.03 | **2.12** | 2.31 |
| | 8K | 2.70 | 0.58 | **0.63** | 0.66 |
| | 4K | 0.88 | 0.23 | **0.27** | 0.24 |
| Qwen3-14B-Thk. | 16K | 8.70 | 1.76 | **1.84** | 2.05 |
| | 8K | 2.46 | 0.53 | **0.58** | 0.61 |
| | 4K | 0.78 | 0.21 | **0.25** | 0.22 |

across diverse datasets and architectural scaling laws.

## 5.3. Efficiency Evaluation

As shown in Tab. 2, ProphetKV achieves up to a $5\times$ speedup over full recomputation at a $20\%$ recomputation ratio for both 8K and 16K contexts, where the computation is dominated by attention over long sequences. For 4K contexts, however, such speedups are not observed for all methods, as fixed system overheads (e.g., kernel launch and cache management) become more pronounced and limit the achievable acceleration. Compared to existing baselines, ProphetKV matches EPIC in efficiency and outperforms CacheBlend, thanks to its lightweight first-stage user-query-to-context attention, in contrast to CacheBlend's heavier first-layer selection. Consequently, ProphetKV offers a superior trade-off: it matches the speed of the fastest baselines while delivering significantly higher accuracy. A detailed latency breakdown in Appendix E.2 further shows that its newly introduced query-scoring stage is nearly negligible in long-context settings.

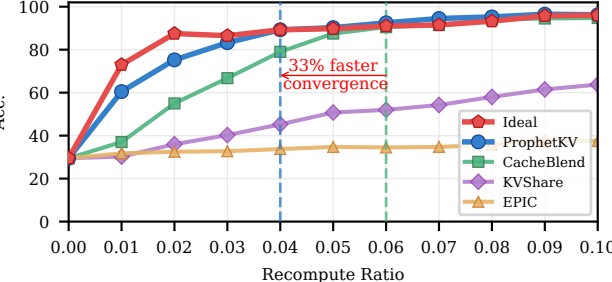

*Figure 9.* Accuracy of the idealized selection strategy, evaluated on the RULER-MV dataset. The ideal refers to the Eq.5.

## 5.4. Ablation Study

**Ablation Study on Selection Strategy.** We first investigate the effectiveness of different token selection strategies under an idealized evaluation setting, aiming to isolate performance degradation arising from selection-quality approximations, such as using first-layer information to predict deeper-layer importance. Specifically, we independently execute the full prefill pipeline and the naïve reuse pipeline to collect the exact oracle information required by each method. Using this oracle information, we then apply a unified partial recomputation pipeline—identical to Stage II of ProphetKV (see Sec. 4.3)—across all methods. Detailed implementation procedures are provided in Appendix C.2.

As illustrated in Fig. 9, the ideal value function (Eq. 5) converges rapidly with a recomputation ratio of only 0.02. ProphetKV closely follows this ideal behavior, achieving convergence with a recomputation ratio of 0.04, which is the fastest among all practical methods. In contrast, the best prior method, CacheBlend, requires a recomputation ratio of approximately 0.06 to reach a similar level of overlap. Compared with CacheBlend, ProphetKV achieves approximately a 33% improvement in convergence speed. These results

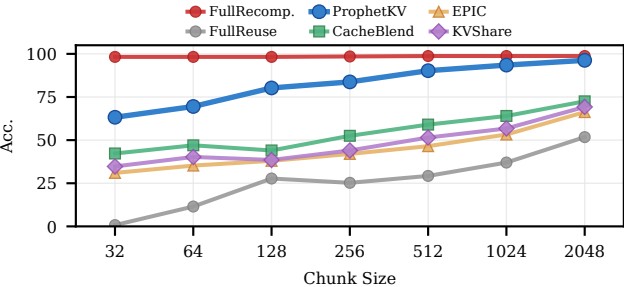

*Figure 10.* Impact of chunk size on accuracy, evaluated on RULER-MV dataset. The recomputation budget is limited to 20%.

support our intuition that ProphetKV's user-query-based selection strategy is more effective than prior approaches, even under an idealized setting simulating unlimited computational resources for token selection.

**Ablation Study on Recompute Ratio.** We analyze the robustness of ProphetKV by sweeping the recomputation ratio from 0.0 to 1.0 on RULER MultiValue (retrieval-intensive) and LongBench Musique (reasoning-intensive). As illustrated in Fig. 8, ProphetKV exhibits a significantly steeper accuracy recovery curve than the baselines. It incurs less than a 5% accuracy drop with only 10%–30% recomputation, whereas baselines typically require 40%–80% to achieve similar accuracy. Notably, on RULER-MV, ProphetKV attains near-complete accuracy recovery at a recomputation ratio of 0.2, while baseline methods exhibit substantially slower accuracy recovery as the recomputation ratio increases. These results confirm that ProphetKV effectively prioritizes the most influential tokens, making it highly robust under constrained computational budgets.

**Ablation Study on Chunk Size.** We further evaluate the impact of chunk size on the accuracy of ProphetKV. As illustrated in Fig. 10, ProphetKV consistently achieves the highest accuracy across all evaluated chunk sizes. In particular, it achieves near-lossless accuracy with chunk sizes above 512 tokens, a common setting in RAG scenarios (Hu et al., 2024). Accuracy for all methods degrades as chunk size decreases, since smaller chunks lead to more missing cross-chunk attention. Therefore, while ProphetKV remains robust under moderate fragmentation, extremely small chunks and very low recomputation ratios remain challenging because the imprecise reused cache provides a weaker proxy for selecting cross-chunk dependencies. Additional selection-overlap results under small chunks are provided in Appendix D.2.

## 6. Related Works and Discussion

**Contemporary query-aware strategies.** Several recent works have identified the limitation of query-agnostic token selection. CacheClip (Yang et al., 2025b) and A3 (Zhou et al., 2025) incorporate query-aware but single-layer signals for refining or recomputing reused KV caches, while

SnapKV (Li et al., 2024) leverages query-side attention patterns for KV cache compression within a single long-context sequence. ProphetKV shares the high-level motivation that the user query provides a useful signal for token importance, but targets a different setting: training-free, position-independent KV cache reuse over independently precomputed RAG chunks, where token importance must be fused across Transformer layers to produce a stable recomputation set. We provide a detailed comparison and layer-policy ablation in Appendix C.3.

**Discussion on Multi-Turn Usage.** Multi-turn interactions can be viewed as sequential single-turn requests with a shared prefix and a dynamic query at each turn, as in systems such as SGLang (Zheng et al., 2024). ProphetKV naturally fits this setting: at each turn, Stage I re-evaluates token importance with respect to the current query, so the recomputation set can adapt as the user's intent shifts and mitigate multi-turn misalignment.

**Limitations and Future Work.** ProphetKV is an algorithmic framework for query-aware partial recomputation, and our current implementation is not yet fully integrated into production inference engines with optimized kernels, batching, and scheduling policies. Such system-level integration is orthogonal to the selection strategy, but it can affect end-to-end latency and throughput in real deployments.

Moreover, ProphetKV does not address retrieval quality itself. Noisy, contradictory, or adversarially retrieved chunks may distract the query-attention signal. Future work could combine ProphetKV with query rewriting, retrieval-quality-aware scoring, or lightweight compensation mechanisms to recover such indirectly useful evidence.

## 7. Conclusion

We proposed ProphetKV, a high-fidelity, position-independent KV cache reuse mechanism for long-context RAG scenarios. It leverages query-driven selective recomputation to recover task-critical cross-attention and mitigates the accuracy loss observed in prior work with minimal overhead. Extensive experiments show that ProphetKV significantly improves accuracy compared to SOTA approaches.

## Acknowledgements

We thank Beijing Yanrong Technology Co., Ltd. for providing support and research funding. This work was partially supported by the National Natural Science Foundation of China (Grant Nos. 62502119 and 62472127), the Shenzhen Science and Technology Program (Grant No. GXWD20231128111309001), and the Guangdong Basic and Applied Basic Research Foundation (Grant No. 2023A1515110072).

## Impact Statement

This paper presents work whose goal is to advance the field of machine learning. There are many potential societal consequences of our work, none of which we feel must be specifically highlighted here.

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

# A. Pseudocode of Our Method

Algorithm 1 presents the dual-stage partial recomputation procedure adopted in ProphetKV. The algorithm takes the precomputed KV caches from multiple chunks across all layers, along with the user query tokens $Q_s$, as input and aims to selectively recover task-critical cross-attention semantics with minimal recomputation cost. In the first stage, the algorithm computes a user-query-aware importance score for each context token at each layer by measuring the attention induced by the user query, resulting in a layer-specific value function $\alpha_l(t)$.

Since attention patterns can vary significantly across layers, directly selecting recomputation tokens independently at each layer leads to unstable and inefficient behavior. To address this issue, the second stage aggregates the layer-wise importance scores into a fused score $\bar{\alpha}(t)$, from which a unified set of top-$p$ tokens is selected for recomputation. Given this unified token set, the KV cache is recomputed independently at each layer, while the remaining cached KV entries are reused directly.

---

**Algorithm 1** Dual-Stage Partial Recomputation Algorithm

---

**Input:** Precomputed KV cache $\{K'^{(l)}_{1:C_1}, V'^{(l)}_{1:C_1}\}^L_{l=1}, \{K'^{(l)}_{1:C_2}, V'^{(l)}_{1:C_2}\}^L_{l=1}, \cdots, \{K'^{(l)}_{1:C_n}, V'^{(l)}_{1:C_n}\}^L_{l=1}$ (multiple chunks and layers), user query tokens $Q_s$.

*// First Stage: User-Query-Aware Token Selection*

**for** $l = 1$ **to** $L$ **do**

    Concatenate all chunks $K'^{(l)}_{1:C} = [K'^{(l)}_{1:C_1}, K'^{(l)}_{1:C_2}, \cdots, K'^{(l)}_{1:C_n}]$ and $V'^{(l)}_{1:C} = [V'^{(l)}_{1:C_1}, V'^{(l)}_{1:C_2}, \cdots, V'^{(l)}_{1:C_n}]$.

    Compute value function $\alpha_l(t) = \hat{\Phi}^{(l)}_{Q_s,t} = \textbf{Softmax}(\frac{Q_s \cdot K'^{(l)}_{1:C}}{\sqrt{d_k}})$ for user query tokens at layer $l$

**end for**

*// Second Stage: Multi-Layer Attention Fusion*

Fuse value function $\bar{\alpha}(t) \leftarrow \sum^L_{l=1} \alpha_l(t)$

Select token indices $T_p \leftarrow \text{Top-}p_{t \leq C} \, \bar{\alpha}(t)$

*// Layer-wise KV Cache Recomputation (Independent across layers)*

**for** $l = 1$ **to** $L$ **do**

    Recompute KV cache $\hat{K}^{(l)}_{T_p}, \hat{V}^{(l)}_{T_p}$ at layer $l$

    Update layer-wise KV cache:

        $\hat{K}^{(l)}_{1:C} \leftarrow K'^{(l)}_{[1:C] \setminus T_p} \cup \hat{K}^{(l)}_{T_p}$

        $\hat{V}^{(l)}_{1:C} \leftarrow V'^{(l)}_{[1:C] \setminus T_p} \cup \hat{V}^{(l)}_{T_p}$

    Calculate user query KV Cache: $\hat{K}^{(l)}_{Q_s}, \hat{V}^{(l)}_{Q_s}$

**end for**

**Output:** Reconstructed KV cache $\{\hat{K}^{(l)}_{1:C}, \hat{V}^{(l)}_{1:C}\}^L_{l=1}$, user query KV Cache $\{\hat{K}^{(l)}_{Q_s}, \hat{V}^{(l)}_{Q_s}\}^L_{l=1}$

---

# B. Prompt and Layout Robustness

We provide the full prompt of the example in Sec. 3.1 in the following, sentences related to the user query are underlined.

---

**Chunk 1(32 Tokens):** A new coffee shop opened near the central park last week. John's house is in London. The city library extended its opening hours to 9 PM daily.

**Chunk 2(34 Tokens):** The summer music festival attracted over ten thousand attendees this year. Alice's house is in Paris. A new batch of public bicycles was put into use in the urban area.

**Chunk 3(33 Tokens):** The downtown art gallery is hosting a modern painting exhibition this week. Alice stays in John's house on Monday. The local football team won the regional championship last month.

**User Query:** In which city does Alice stay on Monday?

---

*Table 3.* Evaluation on Prompt-order robustness. Context–Query follows the standard RAG layout where the scoring query can attend to the retrieved context. Query–Context places the query before the retrieved context, weakening Stage I under causal masking.

| Prompt order | Model | MQue | HQA | WQA | RULER-VT | RULER-QA1 |
|---|---|---|---|---|---|---|
| Query→Context | Llama3.1-8B | 15.80 | 35.30 | 27.20 | 62.00 | 28.33 |
| Context→Query | Llama3.1-8B | **28.60** | **54.00** | **44.00** | **82.20** | **72.80** |
| Query→Context | Qwen2.5-14B | **34.70** | 60.20 | 52.50 | **100.00** | 39.10 |
| Context→Query | Qwen2.5-14B | 34.50 | **64.30** | **56.90** | 94.60 | **71.20** |
| Query→Context | Qwen3-14B | **35.00** | 58.50 | 63.70 | **100.00** | 58.08 |
| Context→Query | Qwen3-14B | 34.90 | **59.50** | **66.80** | 99.60 | **73.08** |

> **prefix:** Answer the question based on the given passages. Only give me the answer and do not output any other words.\n\nThe following are given passages.\n
> **query:** \n\nAnswer the question based on the given passages. Only give me the answer and do not output any other words.\n\nQuestion:
> **suffix:** \nAnswer:

We merge the above prompt components into the final input prompt as follows:

> {prefix}\n\n{Chunk 1}\n\n{Chunk 2}\n\n{Chunk 3}\n\n{query}{User Query}{suffix}

Following a typical LLM inference workflow, we pass the merged prompt to the model for answer generation. The selected token subsets from different methods are exported and highlighted in red in Fig. 2.

### B.1. Prompt Layout Robustness

ProphetKV uses the user query tokens as the Stage I scoring tokens. This design requires the scoring query to be able to attend to the retrieved context. The standard RAG layout satisfies this requirement by placing the context before the final user query or instruction. Under causal masking, this layout lets the terminal query aggregate evidence from all preceding chunks and yields a useful query-to-context attention signal. In contrast, if the user query is placed before the context, the leading query tokens cannot attend to later evidence, so the Stage I signal becomes weaker. Therefore, we treat query-at-end as a practical serving requirement rather than claiming that prompt order has no effect. To quantify this effect, we compare the standard Context→Query layout with an inverted Query→Context layout under the full prefill setting.

As shown in Tab. 3, the standard context-before-query layout generally yields stronger performance, especially on Llama3.1-8B and RULER-QA1. This supports the mechanism above: placing the query after the context gives Stage I access to the evidence tokens that should be scored for recomputation. The inverted layout can remain competitive on some tasks and models, but it is less stable because its query tokens cannot directly attend to subsequent context. These results suggest that ProphetKV is compatible with common RAG templates, while prompt order should still be treated as an explicit deployment condition.

## C. Selection and Layer-Fusion Analysis

This section provides additional details for the token-selection design. We first compare the value functions used by prior recomputation methods, then discuss related query-aware methods, and finally analyze why selecting from a single fixed layer is brittle.

### C.1. Comparison of Selection Strategies of Prior Works

We compare the selection strategies of prior works, including CacheBlend (Yao et al., 2025), KVShare (Yang et al., 2025c), and EPIC (Hu et al., 2024). Specifically, we adopt the same notation as in Sec. 4.2.

Notably, from Fig. 9, the value function of ProphetKV achieves the fastest convergence speed among all the above methods in an idealized setting without any approximation error, which validates the effectiveness of our selection strategy.

*Table 4.* Comparison of selection strategies of prior works.

| Method | Value Function($\alpha(t)$) | Category | Used low-layer approximation |
|---|---|---|---|
| EPIC | -(token distance to chunk start) | Static | No |
| CacheBlend | $\|\Delta V_t\|_2$ | Dynamic | Yes |
| KV Share | $(\sum_{1 \leq n \leq s} \Phi_{n,t}) \cdot \|\Delta V_t\|_1$ | Dynamic | Yes |
| ProphetKV | $\hat{\Phi}'_{Q_s,t}$ | Dynamic | No |

## C.2. Idealized Evaluation Setting

First, to compute the ideal selection method, we apply Eq. 5 to each token to obtain an importance score. This requires a full recomputation pass to obtain the accurate Value cache ($V$) and the attention weights from user query tokens to all context tokens ($\hat{\Phi}_{Q_s,t}$). We apply a mean aggregation over the batch and head dimensions, resulting in an attention weight matrix of shape [Q_tokens, cached_tokens] and a Value matrix of shape [cached_tokens, hidden_dim] for each layer. We then perform column-wise aggregation along the query-token dimension to obtain an attention weights vector for each context token. Finally, following the layer-wise structure of the Transformer, we aggregate the Value matrices and attention weight vectors across all layers to produce the final $V$ and $\hat{\Phi}_{Q_s,t}$.

Second, we compute the error terms involving $V'$ and $\hat{\Phi}'_{Q_s,t}$ via a reuse evaluation pass, using the same dimension-reduction procedure as above. We compute $\Delta\hat{\Phi}_{Q_s,t}$ as the absolute deviation between $\hat{\Phi}_{Q_s,t}$ and $\hat{\Phi}'_{Q_s,t}$ along the token dimension. Similarly, $\Delta V_t$ is computed as the $\ell_2$ norm of the difference between $V$ and $V'$ for each token along the hidden dimension. Using these quantities, we compute the importance score for each token according to Eq. 5 and select the top-$p$ tokens for recomputation.

Third, we recompute the KV cache for the selected tokens. We follow the same procedure as CacheBlend, with the key difference that we do not truncate the query or replace the selected tokens' key-value cache after the first layer. Instead, after token embedding, we directly prune the hidden states, positional embedding matrix, and attention mask to retain only the selected tokens for recomputation. During the forward pass, we pass the selected token indices to the attention function to specify how the key-value cache should be replaced.

Notably, these three steps can be integrated into a single model forward pass without requiring multi-turn model loading, and can be efficiently implemented in frameworks such as PyTorch or TensorFlow. We abstract the layer-wise computation into a reusable function and invoke it three times with different inputs to obtain the variables required for ideal selection and recomputation. Special care must be taken when manipulating positional embeddings to ensure that the existing Key cache is correctly aligned with the newly generated cache, as misalignment can lead to erroneous cache replacement. In particular, positional embeddings should be applied to the old Key cache only once, before cache replacement.

## C.3. Comparison with Contemporary Query-Aware Methods

Several recent methods also use query-side signals for KV-cache-based inference. ProphetKV shares this broad motivation, but differs in both the target setting and the layer policy: it addresses position-independent KV cache reuse for independently precomputed RAG chunks, and it fuses token-importance signals across Transformer layers rather than relying on a single-layer proxy.

**Comparison with SnapKV.** SnapKV (Li et al., 2024) compresses the KV cache within a single long-context sequence. It uses an observation window near the prompt tail as a proxy for future generation attention, which is suitable when all context tokens are processed together in one prefill. ProphetKV targets a different setting: retrieved chunks are precomputed independently and later assembled for a user query, so the key issue is repairing missing cross-chunk attention. It therefore uses the actual RAG query as the semantic signal for recomputation and applies layer-wise scoring to handle inter-layer variation.

**Comparison with CacheClip and A3.** CacheClip (Yang et al., 2025b) and A3 (Zhou et al., 2025) are closer to ProphetKV because they also study query-aware KV reuse. CacheClip relies on a fine-tuned auxiliary predictor, whereas ProphetKV is training-free and derives token-importance scores directly from query-to-context attention in the target model. A3 also uses query-driven attention, but adopts a first-layer approximation for recomputation selection; CacheClip instead relies on last-layer information. ProphetKV fuses signals across layers because the optimal recomputation subset can shift

*Table 5.* Comparison of single-layer and average-fusion selection policies under the same recomputation setting. First-layer and last-layer represent A3-style and CacheClip-style layer choices, respectively.

| Layer Policy | Model | RULER-MV | RULER-MQ | WQA | HQA | MQue |
|---|---|---|---|---|---|---|
| First (A3) | Llama3.1-8B | 65.50 | 76.25 | 38.50 | 47.80 | 21.40 |
| Last (CacheClip) | Llama3.1-8B | 94.00 | 93.50 | 40.00 | **50.90** | **24.40** |
| Average (Ours) | Llama3.1-8B | **100.00** | **98.50** | **43.21** | 50.70 | **24.40** |
| First (A3) | Qwen2.5-14B | 48.75 | 85.25 | 35.20 | 42.50 | 13.90 |
| Last (CacheClip) | Qwen2.5-14B | 77.50 | 92.75 | 48.80 | 57.10 | 33.10 |
| Average (Ours) | Qwen2.5-14B | **91.25** | **97.00** | **52.30** | **58.10** | **35.60** |
| First (A3) | Qwen3-14B | 55.75 | 78.50 | 46.10 | 46.60 | 13.60 |
| Last (CacheClip) | Qwen3-14B | 77.25 | 91.75 | 67.60 | 66.40 | 36.80 |
| Average (Ours) | Qwen3-14B | **95.75** | **96.50** | **70.80** | **70.80** | **44.10** |

substantially with depth.

To isolate this layer-policy difference, Tab. 5 compares representative first-layer, last-layer, and average-fusion policies under the same recomputation setting. Average fusion is competitive with, and often substantially better than, single-layer choices, supporting the layer-fusion design.

## C.4. More Details on Inter-Layer Differences

In this section, we use the example prompt in Appendix B to illustrate how attention patterns vary across different layers. We run a full prefill pass for Qwen2.5-14B-Instruct (48 layers in total) and extract the normalized attention weights among context tokens at selected layers to visualize the resulting cross-attention patterns, as shown in Fig. 11.

As illustrated in Fig. 11, lower layers (1, 12) focus primarily on nearby tokens, whereas higher layers (26, 36, 48) also attend to distant tokens. These layer-specific patterns explain the low overlap between first-layer attention weights and those of other layers shown in Fig. 7 in the main text.

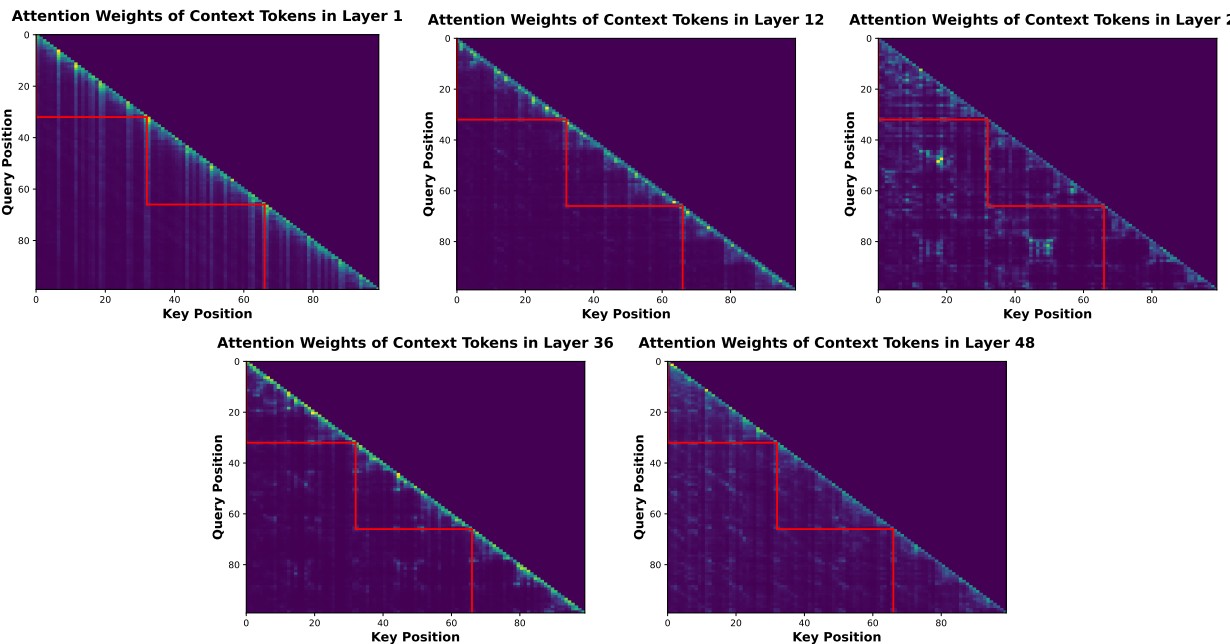

*Figure 11.* Attention weights of context tokens at different layers. The cross-attention region is outlined in red in the bottom-left corner.

## C.5. Layer-Depth Selection Ablation

To further justify the uniform average used in Eq. 7, we compare selection policies that use a single fixed layer at different *relative depths*. Here, relative depth is defined as $d = \ell/L$, where $\ell$ is the selected layer index and $L$ is the total number of

*Table 6.* Layer-depth selection ablation under the same recomputation setting. Relative depth is $d = \ell/L$. Full prefill is included as a reference.

| Layer Policy ($d = \ell/L$) | Model | RULER-MV | RULER-MQ | WQA | HQA | MQue |
|---|---|---|---|---|---|---|
| First layer | Llama3.1-8B | 65.50 | 76.25 | 38.50 | 47.80 | 21.40 |
| 0.25 depth | Llama3.1-8B | 92.00 | 90.25 | 39.30 | 48.00 | 20.50 |
| 0.50 depth | Llama3.1-8B | 99.25 | **98.50** | **44.80** | **52.20** | **26.90** |
| 0.75 depth | Llama3.1-8B | 98.00 | 96.00 | 42.80 | 51.80 | 23.30 |
| Last layer | Llama3.1-8B | 94.00 | 93.50 | 40.00 | 50.90 | 24.40 |
| Average (Ours) | Llama3.1-8B | **100.00** | **98.50** | 43.21 | 50.70 | 24.40 |
| Full prefill | Llama3.1-8B | 100.00 | 100.00 | 43.14 | 53.40 | 26.20 |
| First layer | Qwen3-14B | 55.75 | 78.50 | 46.10 | 46.60 | 13.60 |
| 0.25 depth | Qwen3-14B | 75.50 | 80.75 | 66.60 | 64.50 | 39.70 |
| 0.50 depth | Qwen3-14B | **96.75** | 95.50 | 70.00 | 66.70 | 42.10 |
| 0.75 depth | Qwen3-14B | 95.50 | **98.50** | 70.10 | 68.50 | 39.80 |
| Last layer | Qwen3-14B | 77.25 | 91.75 | 67.60 | 66.40 | 36.80 |
| Average (Ours) | Qwen3-14B | 95.75 | 96.50 | **70.80** | **70.80** | **44.10** |
| Full prefill | Qwen3-14B | 98.00 | 100.00 | 73.00 | 74.20 | 49.10 |

Transformer layers; for example, $d = 0.50$ denotes a middle-layer selector. The experiments use Llama3.1-8B-Instruct and Qwen3-14B under the same recomputation setting as the main evaluation.

As shown in Tab. 6, the best fixed depth varies across both models and tasks. For Llama3.1-8B, the 0.50-depth layer performs best on several LongBench tasks, whereas for Qwen3-14B the best fixed layer shifts across metrics, with 0.50 depth leading on RULER-MV and 0.75 depth leading on RULER-MQ. Since no single relative depth consistently dominates, average fusion serves as a more robust default: it avoids committing to a model- or task-specific layer while retaining signals from the full Transformer depth.

# D. Additional Robustness Results

## D.1. Attention Sink Analysis

Attention sinks often concentrate at chunk-initial tokens and can distort reuse-based attention when independently pre-computed chunks are concatenated. EPIC explicitly targets this phenomenon by statically selecting chunk-initial tokens for recomputation. ProphetKV handles these tokens implicitly and adaptively: if a sink token remains important under the query-specific attention distribution, it receives a high importance score and is selected for recomputation, without hard-coding chunk-initial positions.

To validate this behavior, we measure the cumulative attention weight assigned to chunk-initial tokens after recomputation on the RULER benchmark using Qwen2.5-14B-Instruct. Lower values indicate fewer sink-induced attention artifacts. As shown in Tab. 7, FullReuse produces large chunk-initial attention spikes in middle and deep layers, whereas ProphetKV stays close to Full Prefill and EPIC across all measured layers. This suggests that query-driven selection suppresses sink-induced artifacts while preserving an adaptive, query-specific recomputation policy.

*Table 7.* Total attention weight assigned to chunk-initial tokens after recomputation on RULER using Qwen2.5-14B-Instruct. Lower values indicate fewer sink-induced attention artifacts.

| Layer | Full Prefill | FullReuse | EPIC | ProphetKV |
|---|---|---|---|---|
| L0 | 0.0020 | 0.0015 | 0.0020 | 0.0020 |
| L12 | 0.0018 | 0.6821 | 0.0017 | 0.0018 |
| L24 | 0.0056 | 0.5128 | 0.0076 | 0.0058 |
| L36 | 0.0012 | 0.7492 | 0.0014 | 0.0011 |
| L47 | 0.0016 | 0.2415 | 0.0026 | 0.0014 |

## D.2. Selection Overlap under Small Chunks

To further examine high-fragmentation settings, we measure the overlap between each method's selected subset and the ideal selection(e.g. Eq. 6) subset under small chunk sizes. As shown in Tab. 8, ProphetKV maintains a higher overlap than the baselines for both 64-token and 512-token chunks. The main degradation appears when the chunk size is extremely small and the recomputation ratio is very low, which is consistent with the intuition that heavily fragmented contexts provide fewer reliable cross-chunk signals in the imprecise reused cache.

*Table 8.* Overlap ratio with the ideal selection subset under small chunk sizes and different recomputation ratios.

| Chunk | Top-$p$ | CacheBlend | KVShare | ProphetKV | EPIC |
|---|---|---|---|---|---|
| | 0.01 | 0.30 | 0.32 | **0.44** | 0.22 |
| | 0.03 | 0.42 | 0.22 | **0.71** | 0.55 |
| 64 | 0.05 | 0.50 | 0.25 | **0.73** | 0.43 |
| | 0.07 | 0.59 | 0.28 | **0.77** | 0.41 |
| | 0.09 | 0.60 | 0.33 | **0.79** | 0.38 |
| | 0.01 | 0.42 | 0.46 | **0.69** | 0.38 |
| | 0.03 | 0.56 | 0.50 | **0.75** | 0.26 |
| 512 | 0.05 | 0.57 | 0.51 | **0.77** | 0.26 |
| | 0.07 | 0.58 | 0.53 | **0.79** | 0.28 |
| | 0.09 | 0.58 | 0.55 | **0.79** | 0.29 |

# E. Additional Efficiency and Resource Results

## E.1. Peak Memory Usage

Similar to prior KV-cache methods (Li et al., 2024), peak memory usage mainly arises from the softmax operator in Transformer attention. Full prefill computes attention over the entire prompt, so the attention activation scales as $\mathcal{O}(s^2)$ with the context length $s$. In contrast, Stage I of ProphetKV only computes query-to-context attention for the user query tokens, whose attention activation scales as $\mathcal{O}(|Q_s| \cdot s)$; Stage II then recomputes only the selected token subset rather than the full sequence. We therefore measure the memory timeline of different approaches on Llama3.1-8B-Instruct with a 16K context length. As shown in Tab. 9, ProphetKV peaks at 27.8 GiB, comparable to EPIC (27.7 GiB) and substantially lower than full prefill (51.2 GiB). This indicates that the dual-stage pipeline does not introduce a peak-memory spike beyond standard full prefill.

*Table 9.* GPU memory usage timeline (GiB) at 16K context length on Llama3.1-8B-Instruct. Measurements are sampled every 0.5 seconds.

| Method | 0.0s | 0.5s | 1.0s | 1.5s | 2.0s | 2.5s | 3.0s | 3.5s | 4.0s | 4.5s | 5.0s | 5.5s | 6.0s | 6.5s | 7.0s | 7.5s | 8.0s | 8.5s | 9.0s | 9.5s | 10.0s |
|---|---|---|---|---|---|---|---|---|---|---|---|---|---|---|---|---|---|---|---|---|---|
| Full Prefill | 0.0 | 0.5 | 3.5 | 5.6 | 8.3 | 10.5 | 13.5 | 15.9 | 16.6 | 50.3 | 50.5 | 50.7 | 50.9 | 51.1 | **51.2** | **51.2** | **51.2** | **51.2** | **51.2** | 16.1 | 0.0 |
| EPIC | 0.0 | 0.5 | 3.4 | 5.6 | 8.3 | 10.4 | 13.3 | 15.9 | 16.0 | 16.5 | 17.2 | 18.1 | 27.0 | **27.7** | 19.2 | 19.2 | 0.0 | 0.0 | 0.0 | 0.0 | 0.0 |
| ProphetKV | 0.0 | 0.5 | 3.4 | 5.6 | 8.3 | 10.3 | 13.3 | 15.9 | 15.9 | 16.5 | 17.1 | 17.9 | 19.4 | **27.8** | **27.8** | 18.4 | 0.0 | 0.0 | 0.0 | 0.0 | 0.0 |

## E.2. Latency Breakdown

To explain why query-driven selection adds little latency, we decompose ProphetKV's TTFT into Stage I query-to-context scoring and Stage II selective recomputation. As shown in Tab. 10, Stage I costs only 3.2% of full prefill at 4K context length and drops to 1.0% at 16K, because it scales with query-to-context attention while full prefill grows much faster with total context length. Thus, in long-context settings, the newly introduced Stage I becomes nearly negligible; end-to-end TTFT is dominated by Stage II and is governed primarily by the recomputation ratio.

*Table 10.* Latency breakdown of ProphetKV on Llama3.1-8B-Instruct. Stage I denotes query-to-context scoring, and Stage II denotes selective recomputation.

| Context | Stage I | Stage II | Full Prefill | Stage I / Full |
|---|---|---|---|---|
| 4K | 0.015s | 0.138s | 0.464s | 3.2% |
| 8K | 0.036s | 0.319s | 1.478s | 2.4% |
| 16K | 0.050s | 1.082s | 5.231s | 1.0% |

## E.3. Full TTFT Results

We provide the complete TTFT results across different models and context lengths in Tab. 11. KVShare requires calculating the sum of attention weights as weights for $\Delta V$, which introduces additional computational overhead compared to CacheBlend. Naive Reuse achieves the lowest TTFT since it does not perform any cross-attention recomputation; however, its accuracy is significantly compromised.

*Table 11.* TTFT results across different models and context lengths. Each cell shows TTFT in seconds.

| Model | Context | FullRecomp. | EPIC | ProphetKV | CacheBlend | KVShare | NaiveReuse |
|---|---|---|---|---|---|---|---|
| Llama3.1-8B-Inst. | 16K | 5.23 | 1.08 | **1.13** | 1.29 | 1.32 | 0.09 |
| | 8K | 1.48 | 0.32 | **0.35** | 0.38 | 0.38 | 0.05 |
| | 4K | 0.46 | 0.14 | **0.15** | 0.14 | 0.14 | 0.04 |
| Qwen2.5-14B-Inst. | 16K | 9.94 | 2.03 | **2.12** | 2.31 | 2.37 | 0.14 |
| | 8K | 2.70 | 0.58 | **0.63** | 0.66 | 0.67 | 0.08 |
| | 4K | 0.88 | 0.23 | **0.27** | 0.24 | 0.24 | 0.06 |
| Qwen3-14B-Thk. | 16K | 8.70 | 1.76 | **1.84** | 2.05 | 2.10 | 0.12 |
| | 8K | 2.46 | 0.53 | **0.58** | 0.61 | 0.62 | 0.08 |
| | 4K | 0.78 | 0.21 | **0.25** | 0.22 | 0.22 | 0.06 |

# F. Full Accuracy Results

## F.1. Accuracy Results on Ruler with different context lengths

*Table 12.* Performance comparison of different methods on Ruler dataset with 4k context length.

| Methods | CWE | FWE | MK1 | MQ | MV | S1 | S2 | S3 | QA1 | QA2 | VT | Avg. |
|---|---|---|---|---|---|---|---|---|---|---|---|---|
| *Llama3.1-8B-Inst.* | 99.60 | 85.67 | 100.00 | 99.25 | 100.00 | 100.00 | 100.00 | 99.00 | 82.08 | 50.00 | 99.60 | 92.29 |
| NaiveReuse | 97.80 | 90.67 | 73.00 | 60.25 | 46.05 | 64.00 | 97.98 | 93.00 | 74.42 | 49.00 | 47.80 | 72.18 |
| CacheBlend | 97.80 | 95.00 | 88.00 | 88.75 | 81.05 | **100.00** | **100.00** | **98.00** | 78.33 | 50.00 | 55.60 | 84.78 |
| EPIC | 99.00 | **95.67** | 81.00 | 75.75 | 65.79 | 97.00 | 97.98 | 97.00 | 78.08 | 50.00 | 40.80 | 79.82 |
| KVShare | 98.90 | 93.67 | 87.00 | 87.50 | 75.26 | 98.00 | **100.00** | **98.00** | 72.08 | **53.00** | 42.00 | 82.31 |
| ProphetKV | **99.30** | 95.33 | **100.00** | **99.25** | **99.74** | **100.00** | 98.99 | 92.00 | **82.08** | 51.00 | **92.60** | **91.84** |
| *Qwen2.5-14B-Inst.* | 99.30 | 92.33 | 100.00 | 100.00 | 99.47 | 100.00 | 100.00 | 100.00 | 75.83 | 64.00 | 100.00 | 93.72 |
| NaiveReuse | 98.70 | **99.00** | 77.00 | 72.25 | 36.84 | 98.00 | **100.00** | 95.00 | 73.08 | 51.00 | 39.60 | 76.41 |
| CacheBlend | **99.40** | 97.00 | 95.00 | 94.75 | 64.74 | **100.00** | **100.00** | **100.00** | 74.25 | 58.00 | 48.80 | 84.72 |
| EPIC | 99.20 | 97.00 | 89.00 | 90.75 | 53.68 | **100.00** | **100.00** | **100.00** | 74.42 | 60.00 | 37.20 | 81.93 |
| KVShare | 99.30 | 97.33 | 96.00 | 92.50 | 58.16 | **100.00** | **100.00** | **100.00** | 72.50 | 54.00 | 37.60 | 82.49 |
| ProphetKV | 98.40 | 95.67 | **99.00** | **97.50** | **95.00** | **100.00** | **100.00** | 99.00 | **75.58** | **61.00** | **95.60** | **92.43** |
| *Qwen3-14B-Thk.* | - | - | 100.00 | 100.00 | 98.68 | 100.00 | 98.99 | 100.00 | 79.75 | 71.00 | 100.00 | 94.27 |
| NaiveReuse | - | - | 58.00 | 62.00 | 42.11 | 87.00 | 95.96 | 94.00 | 68.08 | 57.00 | 27.20 | 65.71 |
| CacheBlend | - | - | 82.00 | 81.25 | 62.89 | **100.00** | 98.99 | **100.00** | **82.75** | 64.00 | 65.20 | 81.90 |
| EPIC | - | - | 82.00 | 89.25 | 62.63 | **100.00** | 95.96 | **100.00** | 78.83 | 65.00 | 50.00 | 80.41 |
| KVShare | - | - | 80.00 | 85.50 | 59.47 | **100.00** | 97.98 | **100.00** | 78.83 | 61.00 | 55.00 | 79.75 |
| ProphetKV | - | - | **97.00** | **94.50** | **88.95** | **100.00** | **98.99** | **100.00** | 78.42 | **70.00** | **100.00** | **91.98** |

To further evaluate the robustness of ProphetKV and baseline methods under varying context lengths, we conduct additional accuracy experiments on the RULER dataset with 4K and 16K contexts. These experiments follow the same evaluation protocol as described in the main text, allowing us to systematically assess the impact of context length on retrieval-intensive tasks. As shown in Tab. 12 and Tab. 13, ProphetKV consistently achieves accuracy comparable to full recomputation across all tasks and models, regardless of context length. Notably, the performance gap between ProphetKV and other baselines remains substantial, particularly as context length increases, underscoring ProphetKV's ability to effectively prioritize and recover critical information in long-context scenarios.

**Note:** In Tab. 13, the tokenizer of the Qwen2.5-14B-Instruct model produces longer token sequences for the CWE dataset, resulting in context lengths that exceed the model's maximum limit of 17K tokens. This causes out-of-memory (OOM) errors during evaluation on our 80GB GPU, preventing successful task completion.

## F.2. Accuracy Results on other LongBench datasets

In the LongBench dataset, certain tasks include extended datasets for more challenging evaluation, such as 2wikimqa and passage_retrieval_en. We report results on the extended datasets in Tab. 1 when available, and present the original results in Tab. 14. Results for other extended datasets are shown in Tab. 15.

For challenging cases such as 2wikimqa (WQA) and hotpotqa (HQA), Naive Reuse exhibits a significant accuracy degradation relative to full recomputation. This observation suggests that these tasks require a more comprehensive understanding of global context and cross-chunk interactions. In contrast, ProphetKV achieves performance comparable to full recomputation in these settings, demonstrating its effectiveness in preserving critical information necessary for accurate response generation. Conversely, for simpler cases such as gov_report (GRep), Naive Reuse attains performance on par

*Table 13.* Performance comparison of different methods on Ruler dataset with 16k context length.

| Methods | CWE | FWE | MK1 | MQ | MV | S1 | S2 | S3 | QA1 | QA2 | VT | Avg. |
|---|---|---|---|---|---|---|---|---|---|---|---|---|
| *Llama3.1-8B-Inst.* | 86.00 | 94.00 | 98.00 | 98.50 | 99.75 | 99.00 | 100.00 | 100.00 | 73.83 | 45.00 | 85.60 | 89.06 |
| NaiveReuse | 49.60 | **94.33** | 52.00 | 42.50 | 25.75 | 9.00 | 89.00 | 90.00 | 53.17 | 32.00 | 13.20 | 50.05 |
| CacheBlend | 76.20 | 90.33 | 79.00 | 74.75 | 61.25 | 77.00 | 99.00 | 97.00 | 60.17 | 42.00 | 52.00 | 73.52 |
| EPIC | **82.20** | **94.33** | 69.00 | 64.00 | 36.00 | 25.00 | 98.00 | **98.00** | 64.50 | 41.00 | 41.60 | 64.88 |
| KVShare | 67.80 | 87.33 | 75.00 | 68.50 | 55.25 | 69.00 | **100.00** | 96.00 | 64.42 | 40.00 | 23.40 | 67.88 |
| ProphetKV | 77.00 | 93.33 | **96.00** | **94.50** | **96.25** | **99.00** | 100.00 | 94.00 | **76.50** | **44.00** | 54.40 | **84.09** |
| *Qwen2.5-14B-Inst.* | 0.00 | 94.67 | 100.00 | 99.75 | 94.50 | 100.00 | 100.00 | 100.00 | 70.17 | 56.00 | 99.60 | 83.15 |
| NaiveReuse | **0.00** | 97.67 | 57.00 | 42.00 | 26.00 | 97.00 | 97.00 | 83.00 | 29.75 | 36.00 | 30.20 | 54.15 |
| CacheBlend | **0.00** | 97.33 | 88.00 | 89.25 | 43.25 | **100.00** | 98.00 | 98.00 | 49.42 | 43.00 | 61.40 | 69.79 |
| EPIC | **0.00** | **98.67** | 78.00 | 79.75 | 34.00 | **100.00** | 98.00 | 94.00 | 55.42 | 49.00 | 34.80 | 65.60 |
| KVShare | **0.00** | 94.33 | 78.00 | 83.25 | 43.00 | 99.00 | 97.00 | 97.00 | 52.33 | 48.00 | 36.00 | 66.26 |
| ProphetKV | **0.00** | 98.33 | **98.00** | **97.00** | **86.25** | 99.00 | **99.00** | **100.00** | **67.50** | **57.00** | **88.80** | **80.99** |
| *Qwen3-14B-Thk.* | - | - | 99.00 | 100.00 | 98.75 | 100.00 | 97.00 | 100.00 | 76.83 | 69.00 | 100.00 | 93.40 |
| NaiveReuse | - | - | 25.00 | 10.75 | 12.50 | 11.00 | 43.00 | 33.00 | 12.33 | 22.00 | 4.20 | 19.31 |
| CacheBlend | - | - | 60.00 | 63.25 | 45.00 | **100.00** | 97.00 | 99.00 | 55.75 | 56.00 | 59.20 | 70.58 |
| EPIC | - | - | 52.00 | 57.75 | 38.50 | **100.00** | 92.00 | **100.00** | 60.08 | 56.00 | 45.00 | 66.81 |
| KVShare | - | - | 58.00 | 69.50 | 43.00 | **100.00** | **98.00** | 96.00 | 49.42 | 58.00 | 59.40 | 70.15 |
| ProphetKV | - | - | **97.00** | **97.50** | **93.50** | **100.00** | 96.00 | 98.00 | **78.83** | **73.00** | **100.00** | **92.65** |

*Table 14.* LongBench Results

| Methods | WQA | HQA | SAM | DRead | GRep | LCC | MNews | MFQA_en | MFQA_zh | PassCnt | PRetr_en | Qasper | RepoB | TQA | VC | Avg. |
|---|---|---|---|---|---|---|---|---|---|---|---|---|---|---|---|---|
| *Llama3.1-8B-Inst.* | 42.40 | 51.77 | 8.00 | 23.08 | 28.79 | 16.66 | 23.28 | 55.75 | 61.53 | 2.94 | 99.50 | 38.32 | 16.40 | 46.99 | 11.85 | 35.15 |
| NaiveReuse | 37.06 | 44.62 | 7.46 | 20.66 | 30.89 | 16.70 | **24.77** | 41.90 | 46.21 | **5.93** | 26.50 | 33.64 | 9.74 | **47.53** | 6.90 | 26.70 |
| CacheBlend | 35.24 | 51.23 | 8.19 | **23.82** | **31.24** | 17.82 | 24.12 | 49.37 | 52.70 | 5.15 | 60.00 | 37.72 | 13.70 | 46.18 | 6.21 | 30.85 |
| EPIC | 37.77 | 46.02 | 7.51 | 22.19 | 31.13 | 17.24 | 24.17 | 48.45 | 51.32 | 4.41 | 50.50 | **38.86** | 12.34 | 46.47 | 5.10 | 29.57 |
| KVShare | 36.65 | 48.63 | 8.13 | 21.27 | 30.83 | **17.89** | 23.31 | 48.32 | 53.47 | 5.15 | 48.00 | 37.78 | 13.21 | 46.47 | 8.97 | 29.87 |
| ProphetKV | **42.09** | **52.70** | **9.48** | 21.31 | 29.06 | 17.74 | 23.09 | **53.39** | **55.54** | 2.39 | **96.00** | 36.49 | **14.25** | 46.67 | **16.99** | **34.48** |
| *Qwen2.5-14B-Inst.* | 59.71 | 62.82 | 8.60 | 27.97 | 29.93 | 1.75 | 23.21 | 52.18 | 63.93 | 6.20 | 99.00 | 38.06 | 1.37 | 41.59 | 14.80 | 35.41 |
| NaiveReuse | 23.87 | 15.55 | 8.85 | 15.07 | 28.08 | 1.71 | **24.00** | 36.28 | 30.60 | 2.46 | 17.25 | 32.54 | 4.40 | 13.21 | 14.03 | 17.86 |
| CacheBlend | 47.27 | 52.86 | 8.83 | 21.42 | 29.73 | 1.78 | 23.19 | 48.65 | 53.49 | **5.26** | 60.17 | 38.26 | 3.67 | 28.42 | 14.07 | 29.14 |
| EPIC | 44.19 | 52.21 | 8.47 | 20.85 | 29.75 | 1.50 | 23.60 | **50.70** | 49.30 | 1.64 | 41.75 | 38.43 | 3.09 | 21.29 | 14.57 | 26.76 |
| KVShare | 43.15 | 50.72 | **8.92** | 20.62 | 29.94 | **2.08** | 23.20 | 48.21 | 54.33 | 2.46 | 57.25 | 37.55 | 3.89 | 25.41 | 14.27 | 28.13 |
| ProphetKV | **54.43** | **58.63** | 8.85 | 21.78 | 30.28 | 1.73 | 23.16 | 48.29 | **58.08** | 5.15 | **99.00** | 38.86 | 4.70 | **37.95** | 15.47 | **33.76** |
| *Qwen3-14B-Thk.* | 75.52 | 67.82 | 8.87 | 17.93 | 29.05 | 7.12 | 22.14 | 50.44 | 65.79 | 33.33 | 99.75 | 42.23 | 9.50 | 47.40 | 14.33 | 39.41 |
| NaiveReuse | 32.12 | 22.67 | 7.08 | 5.94 | 17.81 | 1.63 | 21.24 | 30.83 | 31.52 | 0.81 | 13.75 | 32.82 | 1.72 | 36.05 | 1.61 | 17.17 |
| CacheBlend | 65.76 | 59.45 | 8.41 | 15.39 | **29.45** | 4.52 | 22.12 | 42.31 | 58.14 | **3.25** | 66.75 | 40.83 | 3.47 | 47.42 | 2.53 | 31.32 |
| EPIC | 63.80 | 56.84 | **8.93** | 15.01 | 29.34 | 3.25 | **22.21** | **45.17** | **59.10** | **3.25** | 57.83 | 40.42 | 3.48 | 47.20 | 2.24 | 30.54 |
| KVShare | 68.53 | 60.69 | 8.26 | 15.28 | 29.31 | 3.75 | 22.02 | 43.12 | 57.05 | 2.44 | 57.50 | **41.85** | 2.99 | 48.18 | 2.60 | 30.90 |
| ProphetKV | **72.22** | **67.74** | 8.51 | **15.52** | 29.03 | **6.45** | 21.94 | 44.54 | 56.27 | **3.25** | **99.75** | 39.19 | 4.00 | **49.31** | 4.47 | **34.81** |

with full recomputation, indicating that these tasks primarily rely on local context and are less sensitive to cross-chunk interactions. In such scenarios, all partial recomputation methods, including ProphetKV, perform well, highlighting their ability to maintain accuracy while reducing computational overhead. Finally, certain cases in Qwen3-14B-Thk. (e.g., passage_count (PassCnt) and repobench-p (RepoB)) suffer from excessively long thinking generation lengths, causing the model to exceed the maximum length of 4K tokens during answer generation. Consequently, the scores for these cases are substantially lower than those of other datasets.

*Table 15.* LongBench (extended dataset) Results

| Methods | SAM | GRep | LCC | MNews | MFQA_en | PassCnt | Qasper | RepoB | Avg. |
|---|---|---|---|---|---|---|---|---|---|
| *Llama3.1-8B-Inst.* | 7.96 | 28.94 | 16.09 | 23.24 | 55.75 | 2.69 | 39.99 | 16.11 | 23.85 |
| NaiveReuse | 7.50 | **31.67** | 14.34 | 22.81 | 41.90 | 7.07 | 35.72 | 9.02 | 21.25 |
| CacheBlend | 7.79 | 30.95 | 13.64 | **23.97** | 49.37 | **7.12** | 37.77 | 13.39 | 23.00 |
| EPIC | 7.31 | 31.08 | 13.04 | 23.64 | 48.45 | 6.11 | **38.61** | 12.20 | 22.55 |
| KVShare | 7.99 | 30.55 | 14.13 | 20.42 | 48.32 | 6.49 | 37.59 | 11.90 | 22.17 |
| ProphetKV | **9.00** | 29.03 | **16.18** | 23.23 | **53.39** | 4.73 | 37.53 | **14.15** | **23.41** |
| *Qwen2.5-14B-Inst.* | 8.69 | 30.19 | 2.97 | 22.23 | 52.18 | 7.95 | 34.47 | 1.60 | 20.04 |
| NaiveReuse | **8.93** | 28.66 | 2.42 | 21.75 | 36.28 | 6.24 | 28.51 | **5.41** | 17.27 |
| CacheBlend | 8.78 | **30.37** | **3.09** | 21.60 | 48.65 | 4.56 | **35.52** | 3.01 | 19.45 |
| EPIC | 8.77 | 30.11 | 2.61 | 21.68 | **50.70** | 5.92 | 34.53 | 2.54 | **19.61** |
| KVShare | 8.47 | 30.00 | 2.59 | 21.56 | 48.21 | **7.60** | 34.36 | 2.50 | 19.41 |
| ProphetKV | 8.80 | 30.29 | 2.41 | **21.80** | 48.29 | 5.16 | 34.47 | 3.99 | 19.40 |
| *Qwen3-14B-Thk.* | 8.32 | 29.59 | 9.07 | 20.86 | 50.44 | 32.09 | 38.55 | 9.30 | 24.78 |
| NaiveReuse | 7.03 | 21.89 | 1.27 | 16.33 | 30.83 | 3.92 | 31.15 | 1.95 | 14.30 |
| CacheBlend | 8.44 | 29.49 | 2.55 | 20.18 | 42.31 | **11.15** | 36.78 | 3.16 | 19.26 |
| EPIC | 8.29 | 29.52 | 2.61 | 20.22 | **45.17** | 8.78 | 36.90 | 2.08 | 19.20 |
| KVShare | 8.26 | **29.59** | 1.98 | 20.17 | 43.12 | 7.43 | **37.96** | 3.11 | 18.95 |
| ProphetKV | **8.62** | 29.42 | **6.27** | **20.24** | 44.54 | 6.08 | 37.02 | **3.20** | **19.42** |

