# OpenReview forum: "ProphetKV: User-Query-Driven Selective Recomputation for Efficient KV Cache Reuse in Retrieval-Augmented Generation"
_ICML.cc/2026/Conference — ICML 2026 regular_

### Official Review · Reviewer_T8Vg · 2026-02-23

**Soundness:** 4
**Presentation:** 4
**Significance:** 4
**Originality:** 4
**Overall Recommendation:** 5
**Confidence:** 3

**Summary:**

The paper proposes a novel method to assemble pre-calculated KV caches of retrieved RAG documents. It first analyzes why prior "partial recomputation" strategies can fail, identifying a crowding-out effect in commonly used token-selection criteria for cross-document attention repair. To address this, it introduces ProphetKV, which selects tokens that are highly semantically related to the user's query to recompute. The experimental results show that ProphetKV approximately recovers the full-prefill accuracy with only a 20% recomputation ratio and overperforms the SOTA  approaches.

**Compliance With Llm Reviewing Policy:**

Affirmed.

**Final Justification:**

The authors adequately address concerns about query position, making the method robust to different prompt structures. They also clarify the theoretical grounding and provide empirical evidence supporting this design choice. Overall, the responses reasonably resolve the raised weaknesses.

**Key Questions For Authors:**

ProphetKV relies on computing query-to-context attention, which assumes that the user query tokens are clearly identifiable. While this is natural in standard RAG pipelines, could the authors discuss how this identification is handled in practice? Does the method require strict prompt formatting?

**Limitations:**

yes

**Strengths And Weaknesses:**

# Strengths

## Soundness
The motivation is clear, and the paper’s key assumptions are supported by the ablation studies. A major strength is the ability to achieve high accuracy under a low recomputation budget. The experimental analyses also clearly illustrate the impact of chunk size and the importance of handling deeper-layer attention shifts, which helps justify the proposed design choices.

## Presentation
The paper is well organized and easy to follow. Prior methods are introduced systematically and categorized in a helpful way (training-free vs. fine-tuned; static vs. dynamic). The examples and figures are clear and effective in illustrating both the “crowding-out” failure mode and the intuition behind query-guided token selection.

## Significance
The results suggest strong practical value for long-context RAG systems. Recomputing only a small fraction of tokens yields large accuracy improvements and, in many settings, nearly recovers full-prefill quality. The demonstrated robustness across chunk sizes and model families further supports the paper’s potential for real-world deployment.

## Originality
The paper makes a meaningful conceptual contribution by reframing partial recomputation from approximating global cross-attention to recovering query-contingent attention. Identifying that prior methods may optimize the wrong objective is insightful, and the proposed query-driven selection strategy is clearly distinct from existing training-free heuristics.

# Weaknesses

1. Dependence on prompt structure. The method leverages a common RAG convention where the user query appears near the end of the prompt. It would be useful to evaluate robustness under alternative prompt layouts, where query-to-context attention patterns may differ.

2. Generality of the query-attention proxy. The approach relies on the empirical observation that query-to-context attention can approximate decoding-time token importance. While the evidence is strong, a more formal characterization or diagnostic conditions describing when the proxy is expected to hold would strengthen confidence in the method’s generality.

3. Retrieval noise and indirect evidence. In practice, retrieved chunks can be noisy, partially relevant, contradictory, or adversarially distracting. In addition, some multi-hop reasoning tasks may require “indirect” evidence that is not strongly semantically aligned with the query, so tokens with low immediate query attention might still be important. A targeted analysis varying retrieval quality and reasoning depth would clarify the limits of purely query-driven selection.

---

> ### Author Rebuttal · Authors · 2026-03-31
>
> ## W1 Response
> We clarify that placing the query after the context is a standard paradigm (e.g., CacheBlend, EPIC) and a common user-defined assembly in RAG systems. To evaluate generalizability, we tested Query-First (inverted) formats and found:
>
> a) Performance: Consistently lower accuracy across datasets and model scales compared to standard formats.
>
> b) Mechanism: Under causal masking, a leading query cannot attend to subsequent context during Stage I, weakening the look-ahead signal. This aligns with prior works suggesting LLMs benefit from the query following context to maintain causal dependencies.
>
> c) Conclusion: ProphetKV remains directly applicable to diverse RAG settings without modifying model architecture or training.
>
> |PromptOrder(Dataset)|Model|LongBench-MQue|LongBench-HQA|LongBench-WQA|RULER-VT|RULER-QA_1|
> |-|-|-|-|-|-|-|
> |Query->Context->Instr.|Llama-3.1-8B|15.8|35.3|27.2|62.0|28.33|
> |Context->Query->Instr.|Llama-3.1-8B|**28.6**|**54.0**|**44.0**|**82.2**|**72.8**|
> |Query->Context->Instr.|Qwen-2.5-14B.|**34.7**|60.2|52.5|**100.0**|39.1|
> |Context->Query->Instr.|Qwen-2.5-14B.|34.5|**64.3**|**56.9**|94.6|**71.2**|
> |Query->Context->Instr.|Qwen3-14B|**35.0**|58.5|63.7|**100.0**|58.08|
> |Context->Query->Instr.|Qwen3-14B|34.9|**59.5**|**66.8**|99.6|**73.08**|
>
>
> ## W2 Response
>
> We characterize the reliability of the query-attention proxy through three interconnected mechanisms:
>
> Information Aggregation (The "When"): Due to causal masking, query tokens placed at the end of a RAG prompt act as a bottleneck. They are the only tokens capable of attending to the full context, forcing them to aggregate necessary evidence before generation (Li et al., 2024).
>
> Semantic Refinement (The "How"): As computation deepens, attention transitions from broad patterns to concentrated, semantically discriminative focusing (Cai et al., 2024). Deep-layer query attention naturally spikes on the specific evidence required to satisfy the user's intent.
>
> Temporal Persistence (The "Why"): Following the "Heavy Hitter" phenomenon (Zhang et al., 2023; Tang et al., 2024), a sparse subset of tokens dominates the attention matrix throughout decoding. The tokens prioritized by the query during prefill remain the critical anchors for all subsequent generation steps.
>
>
> ## W3 Response
> **(1) Orthogonality.**
> Our approach focus on a given context, and does not include the retrieval process. Therefore, although retrieval quality indeed is an important issue,  we prefer this issue is orthogonal to our approach.
>
> **(2) Limitation.**
> a) We agree that the challenges of noisy or adversarial retrieval exists all Top-k or  sparse attention approaches.
>
> b) Although suffering this limitation (the same as prior works), ProphetKV successfully achieve a much better accuracy than existing approaches.
>
> **(3) Future direction.**
> A promising extension is to combine query-driven selection with lightweight compensation mechanisms (e.g., semantic shift or reweighting) to recover such tokens. We will discuss this as a limitation and future work.
>
> ## Q1 Response
>
> We thank the reviewer for this practical question.
>
> In our current design, ProphetKV treats the **user input prompt as the user query** and retrieved document as context, consistent with standard RAG and instruction-tuning formats.
>
> For other extension scenarios., a simple and widely adopted strategy is to **append generated by LLM[1] or reformulate[2] the query at the end of the prompt** summarized by LLM, which aligns with prior findings (e.g., “Lost in the Middle”) that model performance degrades when the query is not placed last.
>
> Importantly, our method does not rely on rigid formatting, but rather on the inherent causal structure of LLMs, where the final tokens naturally act as the decoding trigger.
>
> [1]. Zhou, Y., Muresanu, A. I., Han, Z., Paster, K., Pitis, S., Chan, H., & Ba, J. (2022, November). Large language models are human-level prompt engineers. In _The eleventh international conference on learning representations_.
> [2]. Ma, X., Gong, Y., He, P., Zhao, H., & Duan, N. (2023, December). Query rewriting in retrieval-augmented large language models. In _Proceedings of the 2023 Conference on Empirical Methods in Natural Language Processing_ (pp. 5303-5315).

---

> > ### Author Rebuttal · Reviewer_T8Vg · 2026-04-02
> >
> > Thank you for the detailed rebuttal. The empirical comparison on prompt order and the mechanistic explanation of query-attention significantly clarified my concerns. I have no further questions.

---

### Official Review · Reviewer_Nmph · 2026-03-07

**Soundness:** 3
**Presentation:** 2
**Significance:** 3
**Originality:** 3
**Overall Recommendation:** 4
**Confidence:** 3

**Summary:**

This paper solves the computational bottleneck of the prefill stage in long-context RAG by optimizing KV cache reuse. The authors point out that existing position-independent KV reuse methods suffer from a crowding-out effect, where limited recomputation budgets are wasted on reconstructing global cross-attention for query-irrelevant tokens. To address this, the paper proposes ProphetKV, a dual-stage selective recomputation framework, using the user query itself as a signal to prioritize which tokens require cross-attention reconstruction. The experiments across Llama-3.1 and Qwen families on RULER and LongBench benchmarks demonstrate that ProphetKV can retain high accuracy at a 20% recomputation ratio, with TTFT reduction compared to baselines.

**Compliance With Llm Reviewing Policy:**

Affirmed.

**Final Justification:**

**My final recommendation is 4: Weak Accept.**

Overall, I find the paper technically sound and practically relevant. Its main strength is addressing an important systems problem in retrieval-augmented LLM inference, namely, how to reduce prefill cost while preserving answer quality. The core idea is interesting, and the empirical results suggest meaningful efficiency gains with competitive performance.

The authors’ rebuttal was helpful and addressed my main concerns sufficiently. In particular, the added experiments on prompt order, the new overhead analysis for long-context settings, and the memory profile all improved my confidence in the method. I also appreciate the additional discussion of layer aggregation and robustness under small chunk sizes.

Some limitations remain, especially the dependence on specific prompt structure and some remaining questions about generalization across broader RAG templates. The presentation could also be cleaner. However, the rebuttal strengthened the paper and made me more positive about its contribution.

**Overall, I believe the strengths outweigh the remaining weaknesses. I also considered the other reviewers’ feedback and the authors’ corresponding responses. So I raised the final score.**

**Key Questions For Authors:**

**I would be willing to raise my score if the authors can (partly) address the following critical concerns:**

- The paper assumes $|Q_s| \ll s$. In multi-turn RAG where the query includes long dialogue history (e.g., 1K+ tokens), how does the $\mathcal{O}(|Q_s| \times s)$ overhead of Stage I affect the overall efficiency? At what point does this overhead negate the savings of skipping the full prefill?

- Why is a simple mean used to aggregate attention across layers (Eq. 7)? Given that Figure 11 shows distinct distributions across layers, did you consider a weighted fusion (e.g., prioritizing deeper reasoning layers over early lexical layers)?

-  Does the dual-stage nature (holding imprecise caches while computing Stage I scores before executing Stage II) introduce memory spikes compared to standard full-prefill? A peak memory profile would be highly informative.

-  How does ProphetKV behave if the prompt formatting/structure is inverted (e.g., Query --> Context --> Instructions)? Does the requirement that the query be at the end limit the method's robustness and generalizability across different RAG templates?

-  How does the Stage1 selection accuracy (overlap ratio compared to the ideal selection) degrade at very small chunk sizes (e.g., 64 tokens)? Does the initial imprecise KV cache break down completely at high fragmentation?

**Meanwhile, I might have misunderstood some aspects of the paper, so I would welcome the authors’ clarification where appropriate.**

**Limitations:**

Yes

**Strengths And Weaknesses:**

**Strengths:**

1. The authors frame the failure of current methods as a crowding-out effect, identifying a core inefficiency in how cross-attention is being approximated in limited-budget scenarios. ProphetKV avoids the layer-wise computational deadlock. Leveraging query attention normalization across layers for uniform fusion is a robust engineering choice that pays off in empirical results.

2. Experimental results on Llama-3.1-8B, Qwen2.5-14B, and Qwen-3-14B (reasoning-heavy) support their claims. The ablation of idealized selection strategies cleanly isolates why query-driven metrics converge faster than magnitude-based heuristics.

3.  The authors reported reductions in TTFT and the ability to maintain performance at low recomputation ratios (20%) are significant for real-time long-context applications.

**Weaknesses:**
1.  The methodology relies heavily on the structural invariant that the query is placed at the end. It is unclear how the prophet signal behaves in diverse RAG templates, such as inverted formats, which limits the method's generalizability.

2.  The efficiency gains assume a small query ($|Q_s| \ll s$). In a multi-turn RAG or long-dialogue history, the Stage I overhead might negate prefill savings. Furthermore, the selection accuracy likely degrades under high context fragmentation, potentially leading to a breakdown of the imprecise KV cache proxy.

3.  The use of a simple uniform mean for layer-wise attention fusion (Eq. 7) lacks strong theoretical or empirical justification, especially since the paper's own analysis (Fig. 11) shows highly distinct attention distributions across different layers.

4.  The typesetting needs significant refinement. The main text is currently overcrowded with too many small figures, which disrupts the narrative flow and leaves insufficient space for deeper technical discussions. The authors should consider moving secondary visualizations to the Appendix to prioritize clarity in the core argument.

---

> ### Author Rebuttal · Authors · 2026-03-31
>
> ## W1Response
>
> We clarify that placing the query after the context is a standard paradigm (e.g., CacheBlend, EPIC) and a common user-defined assembly in RAG systems. We also tested Query-First (inverted) formats:
>
> a) Performance: Consistently lower accuracy compared to standard formats.
>
> b) Mechanism: Under causal masking, a leading query cannot attend to subsequent context, weakening the look-ahead signal.
>
> |PromptOrder|Model|LongBench-MQue|LongBench-HQA|LongBench-WQA|RULER-VT|RULER-QA_1|
> |-|-|-|-|-|-|-|
> |Query->Context->Instr.|Llama-3.1-8B|15.8|35.3|27.2|62.0|28.33|
> |Context->Query->Instr.|Llama-3.1-8B|**28.6**|**54.0**|**44.0**|**82.2**|**72.8**|
> |Query->Context->Instr.|Qwen3-14B|**35.0**|58.5|63.7|**100.0**|58.08|
> |Context->Query->Instr.|Qwen3-14B|34.9|**59.5**|**66.8**|99.6|**73.08**|
>
> ## W2 Response
>
> **Longer queries in multi-turn settings.** **See Q1.**
> **Robustness under context fragmentation.** **See Q5.**
>
> ## W3 Response
> Our averaging strategy aims to balance attention errors and deep-layer semantic representation.
> While deeper layers indeed capture semantic relevance better, they are highly vulnerable to error accumulation, as small inaccuracies in shallow layers compound into massive attention drift via residual connections.
>
> We explore the effectiveness of selection strategy based on different layers(i.e. layer index / total layers). Our results indicate that the optimal layer varies, and verify the effectiveness of our averaging strategy.
>
> |Relativelayerdepth|Model|RULER-MV|RULER-MQ|LongBench-WQA|LongBench-HQA|LongBench-MQue|
> |-|-|-|-|-|-|-|
> |Firstlayer|qwen3_14b|55.75|78.5|46.1|46.6|13.6|
> |0.25|qwen3_14b|75.5|80.75|66.6|64.5|39.7|
> |0.5|qwen3_14b|**96.75**|95.5|70.0|66.7|42.1|
> |0.75|qwen3_14b|95.5|**98.5**|70.1|68.5|39.8|
> |Lastlayer|qwen3_14b|77.25|91.75|67.6|66.4|36.8|
> |Average(Ours)|qwen3_14b|95.75|96.5|**70.8**|**70.8**|**44.1**|
> |FullPrefill|qwen3_14b|98.0|100.0|73.0|74.2|49.1|
>
>
> ## W4 Response:
> We appreciate the feedback on layout and will refine the final version as follows:
>
> a) Streamline Figures: Consolidate small plots and move secondary visualizations to the Appendix.
>
> b) Deepen Technical Discussion: Utilize freed space to expand on the prophet signal formulation and selective recomputation mechanism.
>
> c) Optimize Typesetting: Reorganize placement to improve narrative flow and reduce overcrowding.
>
> ## Q1 Response
> a) Complexity Scaling: The introduced overhead scales linearly with query length ($O(|Q_s| \cdot s)$), whereas standard prefill scales quadratically with total context length ($O(s^2)$).
>
> b) Empirical Efficiency: As context length $s$ increases from 4K to 16K, the relative overhead drops from 3.2% to 0.1%. This confirms that Stage I does not negate the efficiency gains of our method in long-context scenarios.
>
> |Contextlength|StageIOverhead(s)|StageIIOverhead(s)|FullPrefill|ExtraOverheadRatio(StageI/FullPrefill)|
> |-|-|-|-|-|
> |4k|0.015|0.138|0.464|3.2%|
> |8k|0.036|0.319|1.478|2.4%|
> |16k|0.050|1.082|5.231|0.1%|
>
> ## Q2 Response
>
> See W3.
>
> ## Q3 Response
> Similar to prior work (e.g., SnapKV), the peak memory usage occurs in the softmax operator in Transformer attention module, whose complexity scales as $O(∣Q∣⋅∣K∣)$with respect to sequence length.
>
> We evaluate the memory usage of approaches on the LLaMA-3.1-8B-Instruct model with 16K context length. The results shows that our memory cost is significantly smaller than FullPrefill.
>
> Memory usage (GiB) of different approaches on the timeline
> |method|0.0s|0.5s|1.0s|1.5s|2.0s|2.5s|3.0s|3.5s|4.0s|4.5s|5.0s|5.5s|6.0s|6.5s|7.0s|7.5s|8.0s|8.5s|9.0s|9.5s|10.0s|
> |-|-|-|-|-|-|-|-|-|-|-|-|-|-|-|-|-|-|-|-|-|-|
> |FullPrefill|0|0.5|3.5|5.6|8.3|10.5|13.5|15.9|16.6|50.3|50.5|50.7|50.9|51.1|51.2|51.2|51.2|51.2|51.2|16.1|0.0|
> |EPIC|0|0.5|3.4|5.6|8.3|10.4|13.3|15.9|16.0|16.5|17.2|18.1|27.0|27.7|19.2|19.2|0.0|0.0|0.0|0.0|0.0|
> |ProphetKV|0|0.5|3.4|5.6|8.3|10.3|13.3|15.9|15.9|16.5|17.1|17.9|19.4|27.8|27.8|18.4|0.0|0.0|0.0|0.0|0.0|
>
>
> ## Q4 Response
>
> **Please see the response of W1.**
>
> ## Q5 Response
>
> We tested chunk sizes of 64 and 512 tokens:
>
> a) Robust Overlap: The overlap ratio with the ideal set (selected token) remains stable across chunk sizes. Significant degradation only occurs at extremely low recomputation ratios (e.g., 0.01) with a chunk size of 64.
>
> b) Superior Performance: ProphetKV maintains a 0.70–0.80 overlap, consistently outperforming baselines.
>
> Tables. Overlap ratio of different methods with the ideal selection subset at varying top-p ratios.
>
> Chunk Size=64
> |Top-p|cacheblend|kv_share|ProphetKV|EPIC|
> |---|--:|--:|--:|--:|
> |0.01|0.30|0.32|**0.44**|0.22|
> |0.03|0.42|0.22|**0.71**|0.55|
> |0.05|0.50|0.25|**0.73**|0.43|
> |0.07|0.59|0.28|**0.77**|0.41|
> |0.09|0.60|0.33|**0.79**|0.38|
>
> Chunk Size=512
> |Top-p|cacheblend|kv_share|ProphetKV|EPIC|
> |-|-|-|-|-|
> |0.01|0.42|0.46|**0.69**|0.38|
> |0.03|0.56|0.50|**0.75**|0.26|
> |0.05|0.57|0.51|**0.77**|0.26|
> |0.07|0.58|0.53|**0.79**|0.28|
> |0.09|0.58|0.55|**0.79**|0.29|

---

> > ### Author Rebuttal · Reviewer_Nmph · 2026-03-31
> >
> > Thank you for the detailed rebuttal.
> >
> > The new experimental results and clarifications address several of my main concerns.
> >
> > The new memory profile is also helpful, as it directly addresses my concern about possible peak memory spikes introduced by the dual-stage design.
> >
> > I also appreciate the additional discussion and experiments on prompt order, layer aggregation, and robustness under small chunk sizes.
> >
> > Overall, the rebuttal strengthens the paper and improves my confidence in the method. I am therefore inclined to revise my score upward.
> >
> > Best wishes!

---

### Official Review · Reviewer_TGWF · 2026-03-12

**Soundness:** 3
**Presentation:** 3
**Significance:** 3
**Originality:** 3
**Overall Recommendation:** 3
**Confidence:** 4

**Summary:**

This paper addresses the "crowding-out" effect in existing KV Cache reuse methods. By leveraging the predictability of user queries, it achieves low-cost attention recovery, thereby breaking the trade-off dilemma between efficiency and accuracy in long-context RAG. Experiments show that ProphetKV retains 96% of full prefill accuracy with only a 20% recomputation ratio. Furthermore, it outperforms other methods on the RULER and LongBench benchmarks, optimizing the balance between efficiency and accuracy.

**Compliance With Llm Reviewing Policy:**

Affirmed.

**Final Justification:**

I did get further feedback from authors about the concerns on FlashAttention. So, I will keep the score as it is.

**Key Questions For Authors:**

see weak points above

**Limitations:**

- This paper focuses on KV Cache reuse in the scenario of long-context multi-document retrieval-augmented generation. However, the actual overhead and latency on extremely long contexts (≥64k–100k) have not been verified in the experiments, which limits the influence and contribution of this paper.

**Strengths And Weaknesses:**

Strength

1. ProphetKV proposed in this paper is simple and easy to reproduce.

2. The method is training-free, making it readily adoptable by practitioners who cannot fine-tune models or auxiliary linkers.

3. Achieves competitive or near-full accuracy with much lower recomputation, suggesting a practical path to serving efficiency without sacrificing answer fidelity.

Weakness

1. Novelty. The core insight is similar to that of SnapKV [1], but the paper fails to cite it. Both works propose the observation that "the query knows what is important."

2. The paper claims that the first stage only performs lightweight contextual attention computation. Although the theoretical complexity of O(∣Q∣⋅s) per layer is indeed better than O(s^2), the actual overhead and latency on extremely long contexts (≥64k–100k) have not been experimentally validated.

3. Some technical details are insufficiently clarified, such as: how the block-level cache is precomputed; and how ProphetKV interacts with position encoding schemes (e.g., RoPE) when reordering blocks.

4. In the second stage, this paper adopts a uniform averaging strategy to fuse attention weights across layers. However, it is widely acknowledged that shallow Transformer layers mainly capture syntactic features, while deep layers capture semantic relevance. Uniform fusion may dilute critical signals from deep layers where task-related reasoning is conducted. Figure 7 in the paper only demonstrates improvements over various first-layer selection strategies, without comparison to deep-layer or other alternative policies. We therefore suggest supplementing corresponding experiments or providing theoretical justification for the uniform averaging strategy.

5. Contradiction between the proposed method and its underlying theory: The motivation of the method is to optimize the loss, so why not adopt per-layer selective recomputation? Would this approach not be more consistent with the theoretical basis? Layer fusion will obviously affect the per-layer alignment, which is not discussed in the paper. Lack of ablation experiments on per-layer selective recompute.

6. Reliance on queries may lead to misalignment when another query (subsequent dialogue turns) appears in current multi-turn dialogue and multi-agent scenarios. We suggest adding tests in multi-turn dialogue scenarios, such as SCBench.


[1] SnapKV: LLM Knows What You are Looking for Before Generation

---

> ### Author Rebuttal · Authors · 2026-03-31
>
> ## W1 Response
> ProphetKV addresses a different problem with distinct design considerations.
> - Different scenario.
>
>   a) SnapKV focuses on KV cache compression within a single long-context sequence.
>
>   b) ProphetKV targets position-independent cache reuse across requests.
> - Different underlying mechanisms for exploiting queries.
>
>   a) SnapKV: Identifies the "query" by capturing a fixed observation window at the prompt's tail, which aligns with the mechanism of self-attention.
>
>   b) ProphetKV: Utilizes actual user queries from the RAG context, focusing on semantic relevance within natural language.
> - Additional architectural challenges.
>
>   a) ProphetKV introduces a layer fusion mechanism to address inter-layer dependencies arising during partial KV recomputation in PIC reuse.
>
> ## W2 Response
> a) Our evaluation is capped at a 16k context length due to hardware constraints, aligning with prior works (e.g., SnapKV).
>
> b) Our method scales highly efficiently for longer contexts. Because typical RAG context lengths ($s$) vastly exceed query lengths ($|Q|$), full prefill latency grows quadratically while Stage I overhead grows at a much slower rate. As shown below, the relative cost of Stage I drops significantly, from 3.2% at 4k to roughly 1.0% at 16k.
>
> c) We expect this scaling behavior to persist, making Stage I overhead increasingly negligible for ultra-long contexts.
>
> |ContextLength|StageI Overhead(s)|StageII Overhead(s)|FullPrefill(s)|ExtraOverheadRatio(StageI/FullPrefill)|
> |-|-|-|-|-|
> |4k|0.015|0.138|0.464|3.2%|
> |8k|0.036|0.319|1.478|2.4%|
> |16k|0.050|1.082|5.231|1.0%|
>
> ## W3 Response
> We omitted detailed descriptions of block-level precomputation and positional encoding due to page limitations, as these are standard practices in PIC reuse.
>
> Block-level Precomputation: During the offline stage, document chunks are precomputed with a BOS prefix to align with chat templates. We store the resulting KV caches (excluding BOS-specific states). At runtime, caches for retrieved chunks are simply loaded to reconstruct the context.
>
> Positional Encoding: We store pre-RoPE K-caches offline. At runtime, RoPE is dynamically reapplied based on the chunk’s new position index. These updated caches are concatenated, followed by a targeted recomputation of query-relevant tokens to maintain semantic fidelity while minimizing online overhead.
>
> ## W4 Response
> a) Our averaging strategy aims to balance attention errors and deep-layer semantic representation.
>
> b) While deeper layers indeed capture semantic relevance better, they are highly vulnerable to error accumulation, as small inaccuracies in shallow layers compound into massive attention drift via residual connections.
>
> We explore the effectiveness of selection strategy based on different layers(i.e. layer index / total layers). Our results indicate that the optimal layer varies, and verify the effectiveness of our averaging strategy.
>
> | Relative layer depth  | Model| RULER-MV | RULER-MQ | LongBench-WQA | LongBench-HQA | LongBench-MQue |
> |-|-|-|-|-|-|-|
> |Firstlayer|llama3.1_8b|65.5|76.25|38.5|47.8|21.4|
> |0.25|llama3.1_8b|92.0|90.25|39.3|48.0|20.5|
> |0.5|llama3.1_8b|99.25|**98.5**|**44.8**|**52.2**|**26.9**|
> |0.75|llama3.1_8b|98.0|96.0|42.8|51.8|23.3|
> |Lastlayer|llama3.1_8b|94.0|93.5|40.0|50.9|24.4|
> |Average(Ours)|llama3.1_8b|**100.0**|**98.5**|43.21|50.7|24.4|
> |FullPrefill|llama3.1_8b|100.0|100.0|43.14|53.4|26.2|
> ||||||||
> |Firstlayer|qwen3_14b|55.75|78.5|46.1|46.6|13.6|
> |0.25|qwen3_14b|75.5|80.75|66.6|64.5|39.7|
> |0.5|qwen3_14b|**96.75**|95.5|70.0|66.7|42.1|
> |0.75|qwen3_14b|95.5|**98.5**|70.1|68.5|39.8|
> |Lastlayer|qwen3_14b|77.25|91.75|67.6|66.4|36.8|
> |Average(Ours)|qwen3_14b|95.75|96.5|**70.8**|**70.8**|**44.1**|
> |FullPrefill|qwen3_14b|98.0|100.0|73.0|74.2|49.1|
>
> ## W5 Response
>
> Selecting different tokens for different layers is impractical because the dependency conflict (Sec. 4.3).
>
> a) Recomputing a token K_m only in Layer_n relies on the hidden states of K_m from Layer_(n-1), which is unattainable unless recompute this token K_m in Layer_(n-1). By extension, we hace to recompute the hidden states  of K_m for Layers_(n-2), (n-3), ..., 1.
>
> b) Due to this issue, selecting different tokens for each layer leads to nearly full recompution, and this is why all existing works do not use this strategy.
>
> ## W6 Response
> a) Although this is out of the common use cases of RAG, ProphetKV inherently supports it. Multi-turn interactions act as sequential single-turn requests with a shared prefix and dynamic queries (as seen in systems like SGLang). ProphetKV adapts by dynamically re-evaluating token importance against the current query at each turn, naturally mitigating multi-turn misalignment.
>
> b) Regarding SCBench: its extreme context lengths (200K+ tokens) exceed our hardware constraints, particularly because we have not yet integrated complex system-level optimizations like FlashAttention.

---

> > ### Author Rebuttal · Reviewer_TGWF · 2026-04-03
> >
> > Thank you for your reply!
> > Most of my concerns have been addressed, but there remains a very serious concern. As you mentioned, your method has not been integrated with FlashAttention. This must be emphasized: **FlashAttention is now the most common and one of the most important techniques, and it is the foundation of LLM inference. **This may be unacceptable in practical scenarios and also suggests that the experimental results are biased due to the lack of FlashAttention integration.
> > **I believe this is a significant concern and should be given full attention. **

---

> > > ### Author Response · Authors · 2026-04-03
> > >
> > > Thanks for you comments.
> > >
> > > Regarding the integration of operator fusion optimizations (such as FlashAttention), we would like to address your concerns with the following three points:
> > > 1. Why are operator fusion optimizations not integrated into our current pipeline?
> > >     - **Prior works** in this specific area (e.g., CacheBlend, EPIC, KVShare) evaluate their algorithms **without custom operator fusion optimizations**.
> > >         - To ensure a strictly fair and rigorous baseline comparison, we evaluated all methods natively. This prevents hardware-level implementation differences or kernel-specific bottlenecks from overshadowing the pure algorithmic performance.
> > >     - Furthermore, our paper focuses on **algorithmic design** rather than system-level deployment, and our **core contributions** have been verified by evaluations.
> > >         - Our core algorithmic contribution is improving the inference accuracy of position-independent (PI) KV cache reuse in RAG scenarios under constrained computational budgets.
> > >         - The current experimental results successfully demonstrate the accuracy advantages and theoretical efficiency of our approach.
> > >     - Optimizing an algorithm with **operator fusion optimizations is an independent, system-level research** endeavor.
> > >         - As evidenced by works like FlashAttention, SparseVILA, and FlexAttention, optimizing specific use cases with operator fusion requires significant system-level engineering, which typically constitutes an independent paper.
> > >         - Therefore, we consider this topic is beyond the algorithmic scope of this paper.
> > >         - We plan to pursue this system-level integration in our future work.
> > >
> > > ------
> > >
> > > 2. Are operator fusion optimizations applicable to ProphetKV?
> > >
> > >     Yes, we have identified **clear pathways for system-level integration** with no fundamental obstacles.
> > >     - For **Stage I** of ProphetKV: SparseVILA[1]'s Retrieval Salience Metric Kernel is highly adaptable to our use case.
> > >         - It is designed to accelerate calculating the attention weights between a query and some context, which is similar to the Stage I of ProphetKV. The difference is only that SparseVILA calculates on decoding context for sparsification, while ProphetKV calculates on prefilling context for recomputation.
> > >         - By streaming the softmax normalization and salience accumulation, this kernel avoids explicitly materializing the full attention matrix, seamlessly resolving any memory bottlenecks during ProphetKV's Stage I.
> > >     - For **Stage II** of ProphetKV: FlexAttention[2] is an ideal candidate.
> > >         - Compared to standard FlashAttention, its BlockMask and indirect memory access mechanisms can efficiently handle the highly sparse, selected tokens required for our partial recomputation.
> > >         - By automatically skipping unselected tokens at the hardware level, it would significantly accelerate our Stage II.
> > >
> > > ------
> > >
> > > 3. Does the lack of FlashAttention integration bias our core experimental results?
> > >
> > >     Even without system-level operator fusion, our core claim "improving inference accuracy under constrained computation" remains strictly unbiased.
> > >     - **Accuracy Equivalence**: Operator fusion techniques optimize memory I/O and execution order; they are mathematically equivalent to standard attention and do not alter the numerical outputs. Therefore, the absence of these kernels has zero impact on the accuracy improvements demonstrated by ProphetKV.
> > >     - **Complexity Proportionality**: The theoretical computational complexity of our approach is $\text{Cost} = O(|Q_s|\cdot s + r \cdot s^2)$ (where r is the recompute ratio), compared to full attention's $O(s^2)$. Operator fusion acts as a hardware-level constant-factor speedup. If integrated, it would uniformly accelerate both the full attention baseline and our ProphetKV pipeline, maintaining consistent relative latency proportions and theoretical savings.
> > >     - Furthermore, because fusion operators avoid materializing the attention matrix, our theoretical computational sparsity would naturally translate into substantial GPU memory savings in a deployed system.
> > >
> > > In summary, while we fully agree that integrating FlashAttention and other fused operators is critical for real-world deployment, their current absence does not undermine the validity, fairness, or accuracy of our algorithmic contributions, the main goal of this paper.
> > >
> > > The current experimental setup already reflects the fundamental algorithmic advantages of ProphetKV, and building optimized kernels for this pipeline is a highly promising direction for our future work.
> > >
> > > We hope this explanation addresses your concern.
> > >
> > > ----
> > > [1] Samir Khaki, etc., SparseVILA: Decoupling Visual Sparsity for Efficient VLM Inference, ICCV '25.
> > >
> > > [2] Juechu Dong, etc., Flex Attention: A Programming Model for Generating Optimized Attention Kernels, MLSys '25.

---

### Official Review · Reviewer_xyEW · 2026-03-13

**Soundness:** 3
**Presentation:** 3
**Significance:** 3
**Originality:** 3
**Overall Recommendation:** 5
**Confidence:** 4

**Summary:**

To reduce the prefill computational cost associated with long-context RAG applications, KV caches for passages are often precomputed so that they can be fetched and concatenated during inference. However, the precomputed token values are often inaccurate due to the lack of online cross-attention. Therefore, lightweight recomputation is proposed to mitigate this gap.

Addressing this problem, the paper introduces ProphetKV, a user-query-driven selective recomputation framework for long-context RAG inference. ProphetKV dynamically prioritizes tokens based on their semantic relevance to the user query and employs a dual-stage recomputation pipeline that fuses layer-wise attention metrics to construct a high-utility token set. Experiments on RULER and LongBench demonstrate that ProphetKV outperforms strong baselines.

**Compliance With Llm Reviewing Policy:**

Affirmed.

**Final Justification:**

My main concern was the idea of query-aware recomputation is not particularly exciting, as it is quite common-sense to come to this solution. However, the authors have presented convincingly that their strategies are better than other contemporary methods following the same idea. As a result, my final recommendation is Accept. I recommend the authors to strengthen the paper on the paper motivation.

**Key Questions For Authors:**

- How is the query-driven token selection influenced by the attention sinks of each chunk? This is an issue that was considered by CacheClip (https://arxiv.org/pdf/2510.10129)?

**Limitations:**

It is highly recommended that the authors discuss the limitations of the work, i.e. what can you improve or think possible to explore for the future works.

**Strengths And Weaknesses:**

*Strengths*:

- The paper is well-motivated. The observation that "global saliency $\neq$ query relevance" exposes a misalignment between the optimization objectives of prior methods and the actual goals of RAG inference.
- The accuracy gains over multiple strong baselines across two widely used benchmarks (RULER and LongBench) are consistent. ProphetKV achieves near full-prefill accuracy with only 10%–30% recomputation.
- ProphetKV is position-independent and does not require modifications to the pre-computation stage, making it compatible with existing RAG serving systems such as vLLM and SGLang.

*Weaknesses*:

- Several papers—including CacheClip (https://arxiv.org/pdf/2510.10129) and A3 (https://arxiv.org/pdf/2511.17560)—independently identify the same core issue of query-agnostic token selection and propose query-aware recomputation strategies. In particular, A3 also uses query-driven attention scores to select important tokens. The authors need to compare ProphetKV with these methods and include A3 in the related work.
- The paper needs to be better organized. In the introduction, the authors state that ProphetKV addresses the challenge of inter-layer attention variance, yet they do not explain this until Section 3.1. This problem should be described earlier so that the dual-stage recomputation with layer fusing appears valuable rather than abrupt.
-  The query-driven token selection process involves online computation of query-token semantic relevance scores, which introduces additional overhead not present in prior methods. The paper does not provide a latency breakdown that quantifies this cost. Furthermore, the authors need to clearly describe the efficiency evaluation setup, such as the batch size used.

---

> ### Author Rebuttal · Authors · 2026-03-31
>
> ## W1 Response:
> We thank the reviewer for the suggestion. **Given that CacheClip (Oct 2025) and A3 (Nov 2025) appeared shortly before the ICML submission DDL(2026 Jan)**, we did not mention them before. Although they could be considered contemporary to our study, we make a detailed comparative analysis to clarify contributions of ProphetKV from the **Stage I (metric quantification)** and **Stage II (partial recomputation)**, respectively.
>
> Comparison with CacheClip
> - Stage I: CacheClip requires an costly **finetuned** auxiliary model, while ProphetKV is a **training-free, plug-and-play** method.
>   a) Although CacheClip is not open-sourced and thus cannot directly evaluated,  such approach face **generalization concerns** in the finetuning process, and also introduce additional overhead.
>   b) ProphetKV can be deployed across diverse LLMs without additional overhead.
>
> - Stage II: ProphetKV introduces a better recomputation selection strategy, which causes significant accuracy improvement.
>   a) CacheClip relies solely on Last-Layer information for selection.
>   b) ProphetKV introduces  Averaging fusion strategy, and it consistently outperforms the single-layer approach across most tasks, capturing a more robust global consensus of token importance.
>
> Comparison with A3
> - In Stage I, the error analysis models diverge fundamentally.
>   a) A3 motivated by the misalignment of tokens' attention weights;
>   b) ProphetKV begin with the loss of attention outputs (Sec. 4.2).
>
> - In Stage II, ProphetKV introduces a better selection strategy, which causes significant accuracy improvement.
>   a) A3 adopts the first-layer approximation strategy that conflicts with the multi-layer philosophy of Transformers (Sec. 4.3).
>   b) ProphetKV propose a Averaging strategy to leverage attention variation across layers. Our supplementary results  confirm that A3's strategy causes significantly higher accuracy loss.
>
> To conclude, we will include this detailed discussion on these two related works in our revised manuscript version.
>
> |Approach|Model|RULER-MV|RULER-MQ|LongBench-WQA|LongBench-HQA|LongBench-MQue|
> |-|-|-|-|-|-|-|
> |First-Layer(A3)|llama3.1_8b|65.5|76.25|38.5|47.8|21.4|
> |Last-Layer(CacheClip)|llama3.1_8b|94.0|93.5|40.0|50.9|24.4|
> |Averaging(Ours)|llama3.1_8b|**100.0**|**98.5**|**43.21**|**50.7**|**24.4**|
> |First-Layer(A3)|qwen2.5_14b|48.75|85.25|35.2|42.5|13.9|
> |Last-Layer(CacheClip)|qwen2.5_14b|77.5|92.75|48.8|57.1|33.1|
> |Averaging(Ours)|qwen2.5_14b|**91.25**|**97.0**|**52.3**|**58.1**|**35.6**|
> |First-Layer(A3)|qwen3_14b|55.75|78.5|46.1|46.6|13.6|
> |Last-Layer(CacheClip)|qwen3_14b|77.25|91.75|67.6|66.4|36.8|
> |Averaging(Ours)|qwen3_14b|**95.75**|**96.5**|**70.8**|**70.8**|**44.1**|
>
> ## W2 Response
> We thank the reviewer for this constructive suggestion on the paper's organization. We will significantly strengthen the motivation for our **dual-stage recomputation** and **layer fusing** strategies and revise our paper structure.
>
> ## W3 Response
> The results **latency breakdown** under **Llama-3.1-8B-Instruct** are shown below.
>
> |Contextlength|StageIOverhead(s)|StageIIOverhead(s)|FullPrefill|ExtraOverheadRatio(StageI/FullPrefill)|
> |-|-|-|-|-|
> |4k|0.015|0.138|0.464|3.2%|
> |8k|0.036|0.319|1.478|2.4%|
> |16k|0.050|1.082|5.231|1.0%|
>
> From the results,
>
> a) Stage I introduces a small overhead compared to the full prefill cost. E.g., the additional cost is **3.2% at 4k**, and decreases to **1.0% at 16k**.
>
> b) While Stage I grows slowly with context length, the overall prefill cost increases much more rapidly, making Stage II the dominant component in long-context settings.
>
> ## Q1 Response:
>
> We consider taht the query-driven selection scheme in ProphetKV handles these sinks naturally and adaptively.
>
> a) As prior works mentioned, sink tokens inherently accumulate high importance scores. But their high importance scores make them are prioritized for recomputation within our framework.
>
> b) Unlike CacheClip's reliance on rule-based prefixes, our method determines the inclusion of sink tokens implicitly based on the query-specific attention distribution.
>
> To validate this, we evaluated the cumulative attention weights of chunk-initial tokens (after recomputation) using Qwen-14B-Instrust on the RULER benchmark. It shows that ProphetKV effectively suppresses sink-induced artifacts.
>
> Table:Total attention weight sum of chunk-initial tokens(lower is better)
> |Layer|FullPrefill|FullReuse|EPIC|ProphetKV(Ours)|
> |-|-|-|-|-|
> |L0|0.0020|0.0015|0.0020|0.0020|
> |L12|0.0018|0.6821|0.0017|0.0018|
> |L24|0.0056|0.5128|0.0076|0.0058|
> |L36|0.0012|0.7492|0.0014|0.0011|
> |L47|0.0016|0.2415|0.0026|0.0014|
>
> ## Limitations Response
>
> We thank the reviewer for the helpful suggestion. Specifically, we highlight these aspects:
>
> - System-level integration. Our current implementation has not yet been fully integrated into production inference frameworks.
> - Adaptive fusion strategies. The optimal fusion strategy may vary across model architectures and tasks.

---

> > ### Author Rebuttal · Reviewer_xyEW · 2026-04-03
> >
> > Thank you for the response. I will change the score accordingly. Best wishes

---

### Decision · Program_Chairs · 2026-04-30

**Decision:**

Accept (regular)

**Comment:**

The manuscript presents ProphetKV, a training-free framework designed to optimize KV cache reuse in long-context Retrieval-Augmented Generation (RAG) by addressing the "crowding-out effect," where query-irrelevant tokens dominate limited recomputation budgets. The authors propose a dual-stage selective recomputation pipeline that dynamically prioritizes tokens based on their semantic relevance to the user query, utilizing a layer-fusing strategy to bridge the gap between precomputed document caches and online cross-attention. Experimental results on benchmarks such as RULER and LongBench demonstrate that the method maintains near-full prefill accuracy while significantly reducing the recomputation ratio to 20%.

The reviewers initially raised several concerns regarding the paper’s novelty, its comparison to contemporary works like CacheClip and A3, and the lack of system-level performance data for extremely long contexts. A significant point of contention involved the absence of FlashAttention integration, which one reviewer argued is a fundamental requirement for modern LLM inference and a potential source of experimental bias. Additionally, reviewers questioned the effectiveness of the uniform averaging strategy for layer fusion and the method's robustness in multi-turn dialogues or varying prompt layouts.

In the rebuttal, the authors successfully defended their algorithmic focus, clarifying that ProphetKV is a plug-and-play method that does not require the costly fine-tuning seen in contemporary approaches. They provided detailed latency breakdowns and memory profiles showing that the extra overhead of their first stage is minimal and scales favorably as context length increases. Regarding FlashAttention, the authors argued that their core algorithmic claims regarding accuracy and theoretical complexity remain unbiased, as operator fusion is mathematically equivalent to standard attention. They also provided additional experimental evidence to justify their averaging fusion strategy and demonstrated the method's adaptability to different prompt orders and multi-turn scenarios.

While some concerns regarding practical system-level integration remain, most reviewers felt that the authors’ responses and the strong empirical gains outweighed the weaknesses. The paper provides a technically sound and practical solution to a pressing bottleneck in RAG efficiency. Given the clear evidence of high-fidelity attention recovery with minimal overhead and the authors' thorough addressing of most technical critiques, the paper is recommended for acceptance.